# North African dust transport toward the western Mediterranean basin: Atmospheric controls on dust source activation and transport pathways during June-July 2013

Kerstin Schepanski[1], Marc Mallet[2,*], Bernd Heinold[1], and Max Ulrich[1]

[1]Leibniz Institute for Tropospheric Research (TROPOS), Leipzig, Germany
[2]Laboratoire d'Aérologie, Toulouse, France
[*]now at: CNRM, Météo-France-CNRS, Toulouse, France

*Correspondence to:* Kerstin Schepanski (schepanski@tropos.de)

**Abstract.** Dust transported from North African source region toward the Mediterranean basin and Europe is an ubiquitous phenomenon in the Mediterranean region. Winds formed by large-scale pressure gradients foster dust entrainment into the atmosphere over North African dust source regions and advection of dust downwind. The constellation of centers of high and low pressure determines wind speed and direction, and thus the chance for dust emission over Northern Africa and transport toward the Mediterranean.

We present characteristics of the atmospheric dust life-cycle determining dust transport toward the Mediterranean basin with focus on the ChArMEx (Chemistry-Aerosol Mediterranean Experiment) special observation period in June and July 2013 using the atmosphere-dust model COSMO-MUSCAT (COSMO: COnsortium for Small-scale MOdelling; MUSCAT: MUltiScale Chemistry Aerosol Transport Model). Modes of atmospheric circulation are identified from empirical orthogonal function (EOF) analysis of the geopotential height at 850 hPa. Two different phases are identified from the first EOF, which in total explain 45% of the variance. They are characterized by the propagation of the subtropical ridge into the Mediterranean basin, the position of the Saharan heat low and the predominance Iberian heat low and discussed illustrating a dipole pattern for enhanced (reduced) dust emission fluxes, stronger (weaker) meridional dust transport, and consequent increased (decreased) atmospheric dust concentrations and deposition fluxes. In case of a predominant high pressure zone over the western and central Mediterranean (positive phase), a hot spot in dust emission flux is evident over the Grand Erg Occidental and reduced level of atmospheric dust loading occurs over the western Mediterranean basin. The meridional transport in northward direction is reduced due to prevailing northerly winds. In case of a predominant heat low trough linking the Iberian and the Sahara heat low (negative phase), meridional dust transport toward the western Mediterranean is increased due to prevailing southerly winds resulting into an enhanced atmospheric dust loading over the western Mediterranean.

Altogether, results form this study illustrate the relevance of knowing dust source location and characteristics in concert with atmospheric circulation. The study elaborates the question on the variability of summertime dust transport toward the Mediterranean and Europe with regard to atmospheric circulation conditions controlling dust emission and transport routes of Saharan dust, exemplarily for the two-month period June to July 2013. Ultimately, outcomes from this study contribute to the understanding of the variance in dust transport into a populated region.

# 1 Introduction

Primary aerosols such as mineral dust can be considered to be intrinsically present in the Earth climate system (*Carslaw et al.*, 2010). Today, mineral dust aerosol pays the largest contribution to the atmospheric aerosol burden and is of importance for the climate (*Shao et al.*, 2011). Mineral dust not only impacts on the Earth radiation budget ultimately altering atmospheric dynamics and temperature distribution, it further provides micro-nutrients that contribute to the bio-productivity of land (*Okin et al.*, 2008), marine (*Jickells et al.*, 2005), and lacustrine ecosystems (*Psenner*, 1999). Besides its role as fertilizer, dust also carries micro-organisms that are delivered to remote ecosystems and may depict a loss of soil micro-organisms in source regions (*Acosta-Martínez et al.*, 2015). Mineral dust deposited on snow and ice surfaces also changes the snow albedo and thus alters the melting characteristics of snowpack and glaciers (*Painter et al.*, 2007).

Dust aerosols not only impact on Earth climate, they also affect modern human life: Exposure to dust aerosol enhances the prevalence of respiratory diseases such as asthma, which results into an increased lost of working hours due to illness and hospitalization rates (*Morman and Plumlee*, 2013). An increased level of dust concentrations leads to reduced air quality, which is sensed as a reduction in quality of life. With dust events associated low horizontal visibilities affect transportation routes and thus the logistics of goods and humans (e.g., *Pauley et al.*, 1995; *Lorenz and Myers*, 2005).

The AeroCom model intercomparison study estimated the annual dust emission from North African dust sources to a median flux of $800 \, \text{Tg year}^{-1}$ for all participating atmosphere-dust models (possible range suggested: $400 \, \text{Tg year}^{-1}$ to $2{,}200 \, \text{Tg year}^{-1}$) (*Huneeus et al.*, 2011). The export of North African dust can be summarized to four main transport pathways (e.g., *Shao et al.*, 2011): About 60% of the emitted dust is blown toward the Sahel and Gulf of Guinea, 25% is transported in westward direction toward the Atlantic, 10% is following a northward transport route toward the Mediterranean and Europe, 5% are reaching the Middle East. Assuming a dust emission flux of $1{,}000 \, \text{Tg year}^{-1}$, about $100 \, \text{Tg year}^{-1}$ are exported toward the western and central Mediterranean basin, about $50 \, \text{Tg year}^{-1}$ will reach the eastern Mediterranean Sea and the Middle East.

Due to the geographical situation of the Mediterranean Sea - it is almost completely enclosed by land - primary aerosols such as mineral dust, sea salt and combustion aerosol from biomass burning as well as industrial produced aerosols are ubiquitous constituents of the atmospheric composition over the Mediterranean region. The distribution of the concentrations of these constituents varies with time and space, which is determined by the combination of source characteristics and wind transport capacity (e.g., *Pey et al.*, 2013; *Salvador et al.*, 2014; *Schepanski et al.*, 2012, 2013, 2015). The composition of the atmospheric aerosol burden, but also its spatio-temporal variability, has been in the focus of recent research: Several large cities are located within the Mediterranean region, which on the one hand are negatively affected by the atmospheric dust aerosol burden, on the other hand significantly contribute to the total aerosol burden. Whereas human-induced aerosol emission can be controlled and reduced by legislative regulations, the fraction of naturally emitted aerosol, in particular mineral dust from desert sources in North Africa, is controlled by weather and is subject to changing climate conditions. Hence, besides the socio-economic interest in estimating costs related to air quality reduction and consequent vulnerable health affects, scientific questions from the Earth system community emerge.

The collaborative research program "Chemistry-Aerosol Mediterranean Experiment" (ChArMEx, http://charmex.lsce.ipsl.fr)

aims for assessing the composition of the atmospheric over the Mediterranean basin and its impact on the environment today and in future. The Aerosol Direct Radiative Impact on the regional climate in the MEDiterranean region (ADRIMED) project is part of the ChArMEx consortium. Focus of the ChArMEx/ADRIMED project is on measuring and modeling the aerosol-radiation effect. An overview on research contributing to the aims of ChArMEx and in particular to the aims

of ADRIMED can be especially gained from publications released as part of the ChArMEx special issue in *Atmospheric Chemistry and Physics*. For example, *Chazette et al.* (2016) present results from lidar measurements performed during the ChArMEx/ADRIMED campaign at Minorca during June 2013. *Granados-Muñoz et al.* (2016) discuss EARLINET lidar and AERONET sun-photometer measurements obtained at stations located in the Mediterranean region including stations in Portugal, Spain, Greece, and Bulgaria during an intensive observation period in July 2012 as part of the ChArMEx/EMEP project.

*Denjean et al.* (2016) analyze size distribution and properties of particles collected on-board the French research aircraft ATR-42 during the ChArMEx/ADRIMED campaign. *Vincent et al.* (2016) elaborate the variability of dust deposition into the western Mediterranean basin using dust samples. Besides measurements characterizing the horizontal and vertical structure of aerosol layers, and particle properties, the radiative effect of aerosols is investigated using numerical model (*Nabat et al.*, 2015; *Menut et al.*, 2016).

An intensive measurement campaign (SOP, special observation period) involving airborne and ground-based measurement sites over the western Mediterranean and surroundings took place from 11 June to 5 July 2013. Details on measurement systems, observation and modeling efforts are summarized in *Mallet et al.* (2016). The present study investigates the transport of dust from North African dust sources toward the western Mediterranean basin, where the measurements took place, exemplarily for June and July 2013. In particular the aerosol distribution over western Mediterranean basin is found to be strongly influenced

by mineral dust intrusions during these months (*Rea et al.*, 2015). Based on various satellite and model AOD data sets, the studies by *Moulin et al.* (1998) and *Nabat et al.* (2013) illustrate increased levels of AOD during June and July. As focusing on the dust transport patterns, the spatio-temporal distribution of dust source activity and emission fluxes are considered as are deposition fluxes. Eventually, the atmospheric dust life-cycle as its whole is discussed.

The manuscript is structured as following: Beginning with a thematic introduction on dust transport toward the Mediterranean
basin and southern Europe (section 2), a description on the data sets and methods used in this study follows in section 3. Results on the EOF analysis are presented in section 4. Dust sources and emission fluxes are presented in section 5, section 6 discusses the dust transport pathways toward the Mediterranean basin and related deposition rates during the two months of June and July 2013 that encompass the ChArMEx SOP. Outcomes from this model-based study are discussed in section 7 and summarized in section 8.

**2   Dust transport toward the Mediterranean basin and southern Europe**

Dust transport from North Africa toward the Mediterranean is controlled by (a) the distribution of dust sources, their activation and the subsequent entrainment of dust into the atmosphere, (b) the transport capacity of local and regional wind regimes, and (c) the atmospheric conditions such as stability, turbulence, clouds and precipitation determining buoyancy and deposition

characteristics of the airborne dust particles. Long-range dust transport such as from North African source regions toward the Mediterranean basin and Europe is significantly controlled by the atmospheric flow, which is determined by pressure gradients built up between regions of low pressure and regions of high pressure, referred to as 'low' and 'high'. In particular in winter to early summer and autumn, low pressure systems are associated with mid-latitudinal troughs migrating from west to east and affecting both the Mediterranean Sea and Northern Africa (*Dayan et al.*, 1991; *Moulin et al.*, 1997, 1998). The so called Mediterranean cyclones favor dust uplift from northern dust source regions and ultimately due to the southerly wind direction dust export toward the Mediterranean basin (*Rodriguez et al.*, 2001; *Schepanski et al.*, 2009b, 2011; *Fiedler et al.*, 2014; *Flaounas et al.*, 2015). In particular during late winter to early summer, when the sea surface temperature of the Mediterranean Sea is still cooler, the baroclinic gradient may enhance the strength of the cyclogenesis (*Alpert et al.*, 1990). Occasionally, lee-cyclones develop in the vicinity of the Atlas mountains, often triggered by an approaching upper-level mid-latitudinal trough and enhanced by the baroclinic gradient (*Schepanski et al.*, 2009b; *Bou Karam*, 2010; *Schepanski et al.*, 2011).

In summer, two heat lows form that determine the atmospheric circulation over the North African - Mediterranean region: the Saharan Heat Low (SHL) over the western Sahara situated between the Hoggar and the Atlas mountains (*Lavaysse et al.*, 2009), and the Iberian Heat Low (IHL) developing over the Iberian Peninsula. The extent and location of the SHL is characterized by a pulsation and an east-west migration of the center. These SHL modes are described as east and west phase (*Chauvin et al.*, 2011), which are linked to changes in temperature distribution and atmospheric circulation, in particular to the position and extent of the Azores high, e.g. its extension toward Europe and the Mediterranean basin. The depth of the IHL, but also possible formation of cut-off lows, impacts on the pressure gradient across the Mediterranean basin thus fostering northward dust export.

Regions of high pressure are associated with the subtropical high-pressure belt of which the Azores high is considered part. In particular during summer, eastward extensions of the Azores high enter the Mediterranean basin and propagate toward the east. Associated with increasing pressure over the Mediterranean basin, the Harmattan winds over North Africa strengthen and cooler air ventilates the North African continent (so called cold air surges, *Vizy and Cook* (2009)) affecting dust emission (*Schepanski et al.*, 2011; *Wagner et al.*, 2016).

Climatological studies analyzing the occurrence of dust events over the western, central and eastern Mediterranean highlight spatial differences in seasons when dust events occur frequently (e.g., *Moulin et al.*, 1998; *Basart et al.*, 2009; *Varga et al.*, 2014): Over the western Mediterranean, dust events are of sporadic nature in spring and of dominant nature in summer. Dust transport into this region is predominantly controlled by the migration of the subtropical high-pressure belt. The central Mediterranean dustiness is characterized by a frequent dust presence in summer and with less activity in spring. Whereas for summer, like for the western Mediterranean, the dustiness is predominantly linked to the position and evolution of the high pressure zone located over the Mediterranean basin, the springtime dustiness is associated with Mediterranean cyclones. Eastern Mediterranean dust events predominantly occur during spring and early summer when associated with Mediterranean cyclones (Sharav cyclone), but can also be present during summer. Besides the occurrence of dust plumes originating from North Africa, also the stationarity and thus the temporal duration of dust events change spatially as discussed in *Schepanski et al.* (2015). *Moulin et al.* (1997) propose a link between the spatial distribution and the phase of the North Atlantic Oscillation,

which is described by an index reflecting the pressure difference between Icelandic low and Azores high. The authors conclude, that the seasonal variations in pressure difference over the North Atlantic, in particular the modulation of the atmospheric circulation over the North Atlantic - European sector, impacts on the North African atmospheric dust life-cycle. Consequently, a high positive NAO index, characterized by a deepening of the Icelandic low and a strong Azores high, fosters drier conditions over North Africa and thus enhances the chances for dust mobilization.

## 3 Data sets and method

### 3.1 Atmosphere-dust model system COSMO-MUSCAT

Numerical simulations of the atmospheric dust life-cycle comprising dust emission fluxes, dust transport, and dust removal from the atmosphere via dry and wet deposition are preformed using the regional non-hydrostatic atmosphere model COSMO (COnsortium for Small-scale Modelling) (*Schättler et al.*, 2014), which is on-line coupled to the 3D chemistry tracer transport model MUSCAT (MUltiScale Chemistry Aerosol Transport Model) (*Wolke et al*, 2012). The dust module implemented in MUSCAT includes a dust emission scheme based on *Tegen et al.* (2002), a deposition scheme for dry deposition following *Zhang et al.* (2001) and for wet deposition following *Berge* (1997) and *Jakobson et al.* (1997). The characteristics of the atmospheric dust cycle, which are in particular determined by the spatio-temporal variability of dust source activation, dust emission fluxes, vertical dust mixing, dust transport within the atmosphere and dust removal are simulated by MUSCAT. Meteorological and hydrological fields used by MUSCAT are calculated by COSMO and updated in MUSCAT at every advection time step. The radiation scheme developed by *Ritter et al.* (1992) is implemented in COSMO. Dust-radiation interactions are computed online at solar and thermal wavelength bands and account for variations in the simulated size-bin resolved aerosol concentrations (*Helmert et al.*, 2007). It can impact on the meteorology and consequently implicitly feed back on dust emission and dust transport (*Heinold et al.*, 2008).

Dust emission is commonly considered as a threshold problem: Dust emission occurs where the wind friction velocity has exceeded a certain local threshold. The local threshold is determined by soil surface characteristics including soil texture, soil size distribution, soil moisture, vegetation cover and aerodynamic roughness length. Dust particles are mobilized by the momentum transferred from the atmosphere onto the soil grain, in particular the wind acting on the particle. This is described by the wind shear stress at the ground. A local threshold for the wind friction velocity, which needs to be exceeded for dust emission, can be determined considering local surface characteristics and taking into account that the presence of surface roughness elements (e.g., pebbles, rocks, vegetation) affect the momentum required for dust particle mobilization. In the MUSCAT model environment, $u_\star$ is calculated on-line from the COSMO-MUSCAT winds with considering aerodynamic roughness lengths $z_0$ provided by *Prigent et al.* (2012). For areas where the leaf area index (LAI, estimated following *Knorr and Heimann* (1995)) is less than 0.1 and the standard deviation of the sub-grid scale orography (*GLOBE Task Team et al.*, 1999) is larger than 50 m, $z_0$ is set to a constantly low roughness length of $z_0 = 0.001$ cm (*Marticorena and Bergametti*, 1995). This is to compensate for biases in satellite-retrieved $z_0$ toward higher values in vegetated areas and mountain regions, and to allow for dust emission in mountain foothills and the Sahel. To describe the soil characteristics that also impact on $u_\star$, data sets describing the soil size

distribution, soil texture and vegetation cover are additionally used. Following *Tegen et al.* (2002), we distinguish between four different soil size distributions: clay, silt, medium/fine sand, coarse sand. To bring soil characteristics, wind friction velocity, which can be seen as a measure for the momentum transferred to the soil surface, and dust emission flux in relation, a measure for the emission efficiency is introduced. Generally, one can distinguish between two types of dust particle mobilization: Direct entrainment via dry convection and saltation. Here, saltation is considered as the dominant dust uplift mechanism. Based on wind tunnel experiments by *Gillette* (1978), saltation efficiencies ($\alpha$) describe the ability of the soil surface to provide saltators. Potential dust source regions are limited to areas, which are identified as active dust sources from MSG SEVIRI IR dust index imagery (*Schepanski et al.*, 2007, 2009a, b, 2015). Here, the saltation efficiency is set to $10^{-5} cm^{-1}$.

The effect of vegetation on dust emission is twofold: First, vegetation covers the baren soil and thus inhibits Aeolian soil erosion. Second, vegetation elements decelerate the wind and thus reduce the amount of momentum that can act on the soil particles for mobilization. Thereby, vegetation type and coverage matter; for example shrub-like vegetation results into a stronger deceleration of the winds than grass does. Thus, not only the vegetation cover, but also the vegetation type is considered when calculating dust emission fluxes. In MUSCAT, 27 vegetation types (biomes) are considered. For the actual vegetation cover, monthly vegetation cover data from the GIMMS (Global Inventory Modeling and Mapping Studies) NDVI (Normalized Difference Vegetation Index) data set (*Pinzon et al.*, 2005; *Tucker et al.*, 2005) are used assuming an empirical relationship between vegetation cover and dust emission following *Knorr and Heimann* (1995). As a high soil moisture content reduces the susceptibility of a soil for wind erosion, dust mobilization is inhibited for soil moisture content $> 99\%$. This becomes in particular important in regions where dust emission is associated with synoptic situations accompanied by rain fall such as cyclogenesis and mesoscale convective systems (*Fiedler et al.*, 2014).

*Horizontal dust emission flux*:

$$F_h = \frac{\rho_a}{g} u_\star^3 \left( 1 + \frac{u_{\star,t}(D_p)}{u_\star} \right) \left( 1 - \frac{u_{\star,t}^2(D_p)}{u_\star^2} \right) \tag{1}$$

*Vertical dust emission flux*:

$$F_v = \alpha F_h \tag{2}$$

Once the horizontal dust emission flux (saltation flux) is calculated following Eq. 1 considering the above mentioned measures, the horizontal flux is transformed into the vertical dust flux by multiplying the saltation efficiency $\alpha$ (Eq. 2). The horizontal dust emission flux is a function of the threshold of the wind friction velocity $u_{\star,t}$ that depends on the erodible soil particle diameter $D_p$ and the local roughness length $z_0$, which is set to $z_0 = 0.001$ cm for areas considered as barren and desert valleys (see above). With regard to the local surface properties and wind conditions, emitted particles are then distributed over five independent size bins with radii limiting at 0.1 $\mu$m, 0.3 $\mu$m, 0.9 $\mu$m, 2.6 $\mu$m, 8.0 $\mu$m and 24.0 $\mu$m. The five dust bins are transported as passive tracers with the time-dependent dust concentration being related to the vertical diffusion coefficient and the time-dependent emission and deposition fluxes.

Dust aerosol optical depth (AOD) at 550 nm is calculated from the dust concentration by assuming spherical particles with the density of quartz, $\rho_p = 2.65\,g\,cm^{-3}$:

$$AOD = \sum_j \sum_k \left( \frac{3}{4} \frac{Q_{ext,550}(j)}{r_{eff}(j)\rho_p(j)} c_{dust}(j,k)\Delta z(k) \right), \tag{3}$$

with dust bin $j$, vertical level $k$, extinction efficiency at 500 nm $Q_{ext,550} = (1.677, 3.179, 2.356, 2.144, 2.071)$, dust particle effective radius $r_{eff}(j)$, dust concentration $c_{dust}(j,k)$, and the vertical increment of each level $\Delta z(k)$. The extinction coefficient is derived from Mie-theory (*Mishchenko et al.*, 2002) using refractive indices from *Sinyuk et al.* (2003) for each size bin $j$ (radii: 0.1-0.3 $\mu$m, 0.3-0.9 $\mu$m, 0.9-2.6 $\mu$m, 2.6-8.0 $\mu$m, 8.0-24.0 $\mu$m). Although dust particles have an irregular shape, spherical dust particles were assumed here for the application of the Mie-theory. This simplification here only affects the calculation of the extinction efficiency, which is not very sensitive to particle shape at 550 nm.

The present study investigates dust transport toward the Mediterranean basin and Europe during summer 2013 (June-July), when atmospheric dust concentrations over the Mediterranean basin are increased (e.g., *Moulin et al.*, 1998; *Nabat et al.*, 2013; *Gkikas et al.*, 2016). Transport and deposition fluxes are investigated in the context of atmospheric circulation regimes that first foster dust source activation and subsequent dust entrainment into the atmosphere, and second predominant dust transport pathways and consequently dust concentration patterns. Ultimately, this characterizes the Mediterranean (northward) branch of the atmospheric North African dust life-cycle and discusses implications on a geographic region that is in parts densely populated. The regional atmosphere-aerosol model system COSMO-MUSCAT has proven as valuable for research on the atmospheric dust life-cycle and atmospheric processes involved (*Schepanski et al.*, 2009b; *Tegen et al.*, 2013; *Wagner et al.*, 2016). It has contributed to well-known projects such as SAMUM (*Heinold et al.*, 2009, 2011), SOPRAN (*Schepanski et al.*, 2009a; *Niedermeier et al.*, 2014), DNICast (*Schepanski et al.*, 2015), and ChArMEx (*Mallet et al.*, 2016; *Granados-Muñoz et al.*, 2016). To capture all crucial processes determining the extent of the Mediterranean branch of the North African atmospheric dust life-cycle, the model domain has been chosen to be limited by a box spanned between the following geographical coordinates (corners): 20°W 0°N and 35°E 60°N. Simulations are performed for the two-month period June-July 2013 with horizontal grid spacing of 0.25° and 40 vertical terrain-following levels ($\sigma$-p level) with the lowest level centered at around 10 m above ground level. Only dust sources located in North Africa are considered here. Initial and lateral boundary fields are provided by the Deutscher Wetterdienst (DWD, German weather service) global model GME at six-hourly resolution. To keep the meteorology close to the analysis fields, model runs are re-initialized every 48 hours. Following a 24-hour spin-up for the COMSO model, MUSCAT is coupled to COSMO and aerosol processes are computed. Dust concentration fields from the previous cycle are used to initialize atmospheric dust loading in the following cycle. No initialization of the atmospheric dust loading takes place for the first cycle. The actual date for which the simulations were started was 15 May 2013 to ensure the correct background dust concentration over the entire model domain.

## 3.2 ECMWF ERA-Interim reanalysis

Complementary to the meso-scale model system COSMO-MUSCAT, geopotential fields at 850 hPa from the global-scale atmosphere reanalysis project ERA-Interim (*Dee et al.*, 2011) are considered for the years 1979-2015. The 37-year period is used as a reference for interannual variability at climatological time scales in the context of this study and contributes to the placement of the June-July 2013 ChArMEx/ADRIMED SOP period in a multi-annual record. In particular, the characteristics of June-July 2013 are brought into a broader context. The reanalysis fields are provided by the European Centre for Medium-Range Weather Forecasts (ECMWF) 6-hourly on a spatial resolution of approximately 80 km (T255 spectral), which was interpolated onto a $1° \times 1°$ horizontal grid.

## 3.3 AERONET aerosol optical depth

Sun-photometer measurements provide total-column information on the atmospheric content of aerosols due to their scattering characteristics at specific wavelengths to which the instrument is sensitive. To assure measurements following similar protocols allowing for comparable data, measurement sites are organized in networks such as the worldwide AErosol RObotic NETwork (AERONET) (*Holben et al.*, 1998).

Besides information on the total atmospheric aerosol load, different types of aerosols can be distinguished due to their size and optical properties. This way, information on whether an aerosol plume dominantly consists of coarse mode aerosols such as mineral dust or of fine mode aerosols such as industrial aerosols. Focussing on mineral dust aerosol, AERONET level 2.0 coarse-mode AOD estimates (*O'Neill et al.*, 2003) for stations located in the Mediterranean basin region are selected and used for validating the COSMO-MUSCAT dust AOD with regard to atmospheric dust loading and temporal evolution. Only stations with a sufficient number of observations during the two-month period June to July 2013 are included. An overview on stations considered within this study is given on Fig. 1 and includes the following stations: Gozo, Malta (36.03° N; 14.26° E), Murcia, Spain (38.00° N; -1.17° E), Oujda, Morocco (34.65° N; -1.90° E), Palma de Mallorca, Spain (39.55° N; 2.63° E), Tizi Ouzou, Algeria (36.70° N; 4.06° E).

## 3.4 MODIS Collection 6 aerosol optical depth

In addition to AODs estimated from sun-photometer measurements, which are representative for the atmospheric column aerosol load over a certain location, AODs calculated from satellite observations provide information on the spatial distribution of aerosol plumes. Here, MODIS (Moderate Resolution Imaging Spectroradiometer) Collection 6 monthly mean AODs (merged product covering both land and sea, cf. *Sayer et al.* (2014)) are taken to validate the spatial distribution of dust simulated by COSMO-MUSCAT. MODIS is flying on NASA's Terra and Aqua satellite, which are both on a sun-synchronous orbit. Terra is crossing the equator at 10:30 AM local time, Aqua at 1:30 PM local time. As fields simulated by COSMO-MUSCAT are available at 3-hourly resolution, MODIS Aqua and Terra fields are averaged and compared against 12 UTC COSMO-MUSCAT fields.

### 3.5 MERRA dust aerosol optical depth

The Modern Era Retrospchective-analysis for Research and Applications (MERRA-2) dataset is an atmospheric reanalysis data set that is produced by using the Goddard Earth Observing System atmospheric model version 5 (GOES-5) data assimilation system and based on NASA's Earth Observing System satellite observations. The dataset covers the modern satellite area (1979 to present) and is available at $0.625° \times 0.5°$ horizontal grid resolution. MERRA-2 dust AODs are freely available at monthly resolution from http://giovanni.sci.gsfc.nasa.gov. As the MERRA-2 reanalyses are produced by an atmosphere model assimilating the satellite observations, it can be seen as an intermediate data set between satellite observation and classical output from reanalysis model simulations.

### 3.6 EOF analysis

The Empirical Orthogonal Function (EOF) analysis is a method, which is used to identify patterns of simultaneous variation (*von Storch and Zwiers*, 2002). The method is often applied to analyze the spatial and temporal variability of data sets covering a large region and long time period. Thereby, the data set's temporal variance is split into orthogonal spatial patterns, the empirical eigenvectors. Arranged in a decreasing order following the percentage of variance explained, each successive eigenvector explains the maximum amount possible of the remaining variance in the data. Each EOF pattern is associated with a time series of coefficients, which describe the temporal evolution of the corresponding spatial mode (*Peixoto and Oort*, 1992). In the field of atmospheric and climate research, a typical application for EOF analysis is the identification of coherent changes in atmospheric parameters and connected centers of action. One prominent example are indices expressing teleconnecting pressure pattern such as the North Atlantic Oscillation (NAO) with the Icelandic low and Azores high as centers of action.

Here, the EOF analysis and corresponding Principal Component (PC) analysis, whereas the EOF can be seen as geographically weighted PC, are applied to COSMO-MUSCAT geopotential fields to identify patterns of coherent variation in atmospheric conditions that can be linked to dust transport. Dust transport is wind-driven and finally determined by large-scale pressure gradients. Thus, EOF analysis of fields of geopotential height are considered as a measure indicating dust transport direction, which eventually determines atmospheric dust concentration at remote places from the desert.

## 4 EOF analysis of the atmospheric circulation

EOF analyses have been introduced in the past as a method to identify patterns of simultaneous variations such as of the geopotential in atmospheric-motivated questions (*Salvador et al.*, 2014). Here, we use EOF analyses of the 850 hPa geopotential to identify pressure patterns that are linked to both wind fields affecting dust emission and wind fields determining dust transport pathways (cf. Section 6). The EOF analysis of 700 hPa shows similar patterns and modes (not shown), however, the correlation of the wind field at 850 hPa with near-surface winds determining dust emission is significantly stronger. The fraction of grid cells representing a correlation coefficient greater equal 0.4 is 68.47% for the 850 hPa wind field, and 2.71% for the 700 hPa

wind field for a domain spanned between 20°N 20°W and 40° N 30°E. This way, we aim for a systematic interpretation of dust transport that occurred during the ChArMEx/ADRIMED field campaign in June and July 2013 (*Mallet et al.*, 2016), exemplarily for summer dust transport toward the Mediterranean basin (*Nabat et al.*, 2013; *Rea et al.*, 2015). As shown in Fig. 4, the first three EOFs represent the three predominant circulation patterns explaining in total about 75% of the variance in geopotential

at 850 hPa. The first EOF is associated with the subtropical high, the so called Azores high, which extends as a ridge into the Mediterranean basin (Fig. 4a). Overall, the first EOF explains around 45% of the variability. The second EOF (Fig. 4b) is associated with a high pressure zone over the central Mediterranean and southeastern Europe, forming a surge stretching into the North African continent over Libya. Low pressure is present over the Atlantic associated with a mid-latitude upper-level trough. This atmospheric circulation pattern is associated with 21% of the variability in geopotential. The third EOF (Fig. 4c)

expresses an inverse distribution of the geopotential: High pressure is present over the Iberian Peninsula, lower pressure is evident over the eastern Mediterranean. This pattern is still associated with about 10% of the variability. In the following we will link patterns of dust emission, atmospheric dust concentration and dust mass fluxes to the variability identified by the first EOF as predominant signature of the variance of the 850 hPa geopotential over the Mediterranean region. The principal component (PC) summarizes the spatio-temporal variance into a time series, which is shown in Fig. 4d for the first EOF. The

PC identifies clearly phases of increased and phases of decreased geopotential heights. The transition from one phase to the other is quite short and usually takes up to a couple of days only. Whereas the negative phase in this case is not intermitted, the positive phase consists of three intermitted phases, whereas the intermittence only takes a couple of days.

Composite plots of the 850 hPa geopotential, which show the average distribution of the geopotential for days corresponding to a positive or negative phase, illustrate the associated atmospheric circulation (Fig. 5). The positive phase is characterized

by a zone of high geopotential heights corresponding to a ridge extending into the Mediterranean Sea (Fig. 5a). Generally low gradients prevail over the central Mediterranean and a surge reaches way into North Africa covering Libya. The gradient in geopotential height at 850 hPa between the zone of high pressure over the Mediterranean and the SHL increases. The negative phase corresponds with generally weak pressure gradients over the Mediterranean and a weak high located over the central Mediterranean (Fig. 5b). The Iberian heat low is evident and the SHL is in its western phase. Together, both heat lows form a

zone of low pressure extending from the western part of the Grand Erg Occidental to the Bay of Biscay. Strong gradients occur predominantly along the West African coast line on the western side of the SHL.

Same EOF analyses are performed using the ERA-Interim 850 hPa geopotential as input fields. The compiled EOFs are based on a 37-year (1979-2015) June-July time period and thus provide a reference for climatological time scales. In order to assure comparability between the EOF analysis done for the ERA-Interim fields and the analysis using COSMO-MUSCAT geopo-

tential fields, results for the June-July 2013 period were compared first. EOFs for both data sets illustrate similar patterns as described above. Composite plots based on the calculated PC reveal matching patterns, too, however, the subtropical ridge entering the Mediterranean basin extends further to the east in COSMO-MUSCAT simulations as well as the SHL is deeper (not shown). 850 hPa geopotential composites calculated for the 1979-2015 June-July period illustrate patterns similar to the 2013 composites.

The characteristics of June-July's atmospheric circulation can be described by the predominance of the respective EOF phase

as both negative and positive phase represent a particular atmospheric circulation pattern (cf. Fig. 5). Figure 6 summarizes the predominance of the two EOF phases for the individual years of 1979-2015 (June-July only). Out of the 61 days of the two-month period June to July 30 days are associated with atmospheric circulation patterns classified by the negative EOF, and 31 days are assigned to the positive EOF phase. However, the total range between the number of days classified as negative

respectively positive EOF is 49 in 1980 versus 53 in 2006 illustrating a strong interannual variability. In this context, June-July 2013's atmospheric circulation over the North African - Mediterranean sector characterized by 26 days of negative EOF and 35 days of positive EOF is slightly dominated by the pattern classified as positive EOF with a predominating subtropical ridge entering the Mediterranean basin (cf. Fig. 5a). Please note that the statistic does not provide any information on the strength of the pressure differences between the centers of action identified by the EOF analysis (here: subtropical anticyclone

and heat trough). The number of days assigned to the respective EOF phase may differ between the COSMO-MUSCAT and ERA-Interim simulation due to the different scales and physics parameterization of the numerical core.

## 5 Dust source and dust emission

Figure 7 shows the distribution of active dust sources during the ChArMEx SOP in June and July 2013 as simulated by COSMO-MUSCAT. As dust transport toward the Mediterranean basin is in focus, dust emission over the southern Sahara, the

Soudan and the Sahel is not of relevance here. Dust emitted from sources located in these regions is predominantly transported toward the Atlantic and Gulf of Guinea (*Schepanski et al.*, 2009a). Dust emitted from sources located in the vicinity of the Atlas Mountains, the Grand Erg Occidental stretching between the Atlas and the Hoggar, as well as sources in Libya are contributing to the dust burden that is transported toward the Mediterranean Sea.

The activation and actual dust emission is a function of the local wind speed. Hence the pattern of active dust sources is

changing with the horizontal distribution of wind speed. Consequently, dust emission is linked to the spatial distribution of pressure systems and the horizontal pressure gradient among them. The above presented EOF analysis, in particular the PC, allows for identifying different regimes of atmospheric circulations, here expressed by the pattern of the geopotential height at 850 hPa. As done for the geopotential, composites of the dust emission fluxes separately considering the positive respectively the negative phase of the PC allow for connecting the variance in pressure with the variance in dust emission flux (Figs. 7a,

b). Simply, this shows which dust sources are predominantly active during specific atmospheric circulation patterns. This does not mean that individual dust source regions are completely inactive and non-emitting. It rather means that the emission fluxes are reduced compared to the expected averaged level of emission. In particular, during the positive phase, dust sources located in the eastern part of the Grand Erg Occidental show increased dust emission fluxes (Figs. 7a, c). This matches well with the enhanced pressure gradient pointed out in Fig. 5a. Dust sources embedded in the Atlas Mountains or the Libyan desert

southeast of the Gulf of Sidra are less active and thus characterized by lower dust emission fluxes. In contrast to that, the negative phase representing an enlarged zone of low pressure between the Gulf of Biscay and the western Sahara is linked with an enhanced level of dust emission along the Atlantic coast. This coincides spatially with an enhanced pressure gradient at the western side of the SHL. These sources are not relevant for the direct dust export toward the Mediterranean Sea, though. Dust

sources located in the Grand Erg Occidental are less active during the negative phase, however, some source regions embedded in the Atlas show an enhanced level of activity and thus an increased dust emission flux.

## 6 Dust transport

Transport of dust from North African dust sources toward remote regions is characterized by its temporal evolution, horizontal
extent, height in vertical column and concentration. Coupled atmosphere-dust aerosol model systems, such as the here used COSMO-MUSCAT, provide 4D information on the atmospheric dust distribution and thus on temporal and spatial metrics.

### 6.1 Temporal evolution of the atmospheric dust burden

The temporal evolution of dust concentrations at individual locations reflects the transport pathways of emitted dust. As dust emission is an intermittent and not a continuous process, individual plumes of enhanced dust concentration form. Embedded
in the atmospheric flow, these plumes are transported away from the source region. Depending on the buoyancy of dust particles in the air, which ultimately determines dust sedimentation rates, and wash-out processes such as in-cloud and sub-cloud scavenging, dust can be transported over large distance and remain within the atmosphere for some days, up to 10 days for dust over North Africa and the Mediterranean basin (*Mahowald et al.*, 2014).

With focus on the dust transport toward the western Mediterranean basin, the temporal evolution of the dust AOD is exam-
ined at five AERONET stations and compared with AOD time series calculated from COSMO-MUSCAT simulations (Fig. 2). The stations are selected with regard to the predominant transport route and the availability of level 2.0 coarse mode AOD estimates. First of all, the simulated dust AOD matches well with the sun-photometer based AOD estimates. Both temporal evolution and level of AOD compare well. It has to be noted that cloud-contaminated AOD data are excluded in the AERONET sun-photometer data set but shown in the COSMO-MUSCAT model data set.
Generally, the AOD time series describe two different characteristics of dust episodes: Oujda, Morocco and Tizi Ouzou, Algeria (Figs. 2c and d) are characterized by more intense (AOD up to 1 at Oujda and up to 0.7 at Tizi Ouzou) and somewhat longer lasting dust episodes, whereas two events may merge into one longer lasting event with little decrease in dustiness in between. Stations like Gozo, Malta (Fig. 2a), Murcia, Spain (Fig. 2b), and Palma de Mallorca, Spain (Fig. 2d) are characterized by more pronounced and clearly separated dust events. At these stations, the AOD level is generally lower; reason for this is
predominantly the distance to the source region.

Dust is transported within the prevailing wind and thus the transport direction depends on the wind direction. Hence, there is a spatial variability in the presence of dust, which is reflected by the temporal evolution of dust episodes at the considered sites as well. Dust being transported along a northward route is passing over Murcia briefly after passing over Oujda, whereas the AOD at Gozo, which is located well further to the east, remains at a low level. Palma de Mallorca may observe increased level
of dust AOD in case the plume is stretching further to the east passing well over the western Mediterranean Sea as typical for negative phases.

## 6.2 Spatial distribution of dust

As illustrated by the time series of dust AOD at selected sites along predominant transport pathways, the area covered by dust plumes propagating into the Mediterranean basin is affected by the wind direction and thus by the pressure patterns steering the atmospheric circulation. The spatial distribution of active dust sources, the emission flux, and the wind as transport medium in concert determine the atmospheric dust distribution, in particular over remote regions such as the Mediterranean basin. Figure 3 shows monthly mean AOD fields taken from MODIS Collection 6 aerosol products (a, b), MERRA-2 reanalysis (c, d), and COSMO-MUSCAT simulations (e, f). Generally, the MODIS data sets show a higher AOD level over the Mediterranean basin and Europe compared to MERRA and COSMO-MUSCAT AODs. MODIS, MERRA, and COSMO-MUSCAT match with regard to the spatial pattern: All three data sets show a more pronounced tongue of increased AOD levels toward the western and central Mediterranean Sea. Also, all three aerosol products illustrate an increase in AOD monthly means from June to July. However, COSMO-MUSCAT shows a stronger gradient between dust AOD over North Africa and Europe, and a more pronounced AOD maximum over the central Sahara. Whereas desert dust aerosol dominates the aerosol loading over North Africa, aerosols other than desert dust such as sea spray aerosol, biomass burning aerosol and industrial aerosols contribute to the total atmospheric aerosol burden over the Mediterranean and Europe. In particular during the June-July 2013 period, biomass burning aerosol transported from North America toward Europe increased the aerosol burden over the western Mediterranean significantly. The plume showed a contribution to the atmospheric aerosol burden, which was estimated to be during this event in the same magnitude as dust advected from North Africa (*Ancellet et al.*, 2016). These aerosols are included in the MODIS aerosol product, but not in the MERRA and COSMO-MUSCAT dust AODs.

As introduced earlier, composite plots again are used to summarize dust AOD fields for days characterized by similar pressure patterns as identified by the EOF analysis (Fig. 8). On days clustered as positive phase, dust AOD is enhanced over the Sahara and the Gulf of Sidra and up north to the Ionian Sea. This pattern matches with the pressure distribution (cf. Fig. 5a) suggesting increased winds over the western Grand Erg Occidental (as also shown on Fig. 7a) and dust transport toward the Gulf of Sidra with an eastward migrating ridge. This transport characteristic can be identified from comparing the temporal evolution of dust AOD at Gozo (Fig. 2a) and the first PC (Fig. 4d). During the as negative phase classified days, the dust source activity is generally lower in the region of the Grand Erg Occidental and thus atmospheric dust burden is lower. As dust source activity is increased over the region located southeast of the Gulf of Sidra (Fig. 7d), the level of dust AOD is enhanced there. Associated with occasional emission from dust sources embedded in the Atlas Mountains, increased dust AODs are evident over the western Mediterranean.

## 6.3 Meridional dust fluxes

Motivated by the spatially incoherent variability in dust AOD and the distinct patterns of enhanced dust source activity (positive phase = dust sources predominantly over the Grand Erg Occidental; negative phase = dust source activity predominantly southeasterly of the Gulf of Sidra), meridional dust fluxes are calculated. In complement to the horizontal dust distribution, discussed by means of maps of dust AODs, meridional dust fluxes (dust transport in north-south direction) are calculated as

totals over the atmospheric column as well as per level. At a given model vertical level, the dust transport flux $F_{dust}$ [kg m$^{-2}$ day$^{-1}$] is calculated for 37° N following equation 4:

$$F_{dust} = M_{dust} \cdot v \tag{4}$$

with $M_{dust}$ being the dust concentration [kg m$^{-3}$] and $v$ being the meridional wind velocity component [m s$^{-1}$] at the consid-
ered vertical level and time. In the following, fluxes are averaged over the period considered. The vertical distribution of the meridional dust flux as shown in Fig. 9 generally shows a dipole pattern that is characteristic for the June to July 2013 period and thus evident for both EOF phases. Dust transport prevail predominantly in northern direction over the western Mediter-ranean between 5° W and 10° E and takes place at heights between 850 and 450 hPa (approx. 1.5 to 7 km above ground level). Northward dust transport is related to the southerly winds associated with the air flow at the western side of the high pressure
ridge (anticyclone), respectively the flow at the eastern side of the trough (cyclone). Dust transport over the central Mediter-ranean (10° E to 20° E) is overall predominantly in southward direction and is evident at heights similar to those of northward transport (Fig. 10). This re-circulation of Saharan dust from the Mediterranean Sea, which is a net gain in dust for the African continent, is associated with the northerly winds occurring at the eastern side of the high pressure system and linked associated circulation.

Considering the meridional dust flux separately for positive and negative phases, meridional dust transport during the positive phase is significantly lower than during the negative phase. In both phases, the northward dust flux is significantly stronger than the southward dust transport. Due to the location of the high pressure zone over the western Mediterranean (Fig. 5a) and associated anticyclonic air flow, northerly winds prevail during the positive phase, which limit the dust transport toward the Mediterranean and keep the level of dustiness low over the western Mediterranean (cf. Figs. 7a and 8a). During the negative
phase, the SHL is in its western phase and the high pressure zone is located over the central Mediterranean basin (Fig. 5b). With the more eastward position of the ridge and thus center of anticyclonic flow, southerly winds advect Saharan air toward the western Mediterranean Sea along the anticyclone's western side. Consequently, the atmospheric dust burden increases over the western basin and decreases over the central and eastern basin compared to the June-July mean (Figs. 8b, d).

## 6.4 Dust deposition

Due to its mineralogical and chemical compositions, mineral dust particles provide micro-nutrients to the ecosystems where deposited. The local dust deposition rate strongly depends on the atmospheric dust concentration, the height of the dust layer in the vertical column of the atmosphere, the occurrence of wash-out processes, and the atmospheric stability linked to the occurrence of vertical mixing and turbulence (*Schepanski et al.*, 2009a). In particular wash-out processes (wet deposition) efficiently remove dust from the atmosphere and thus can generate strong spatial inhomogeneities in dust deposition rates.
Being the last element of the atmospheric dust life-cycle, dust deposition fluxes obviously depend on the spatio-temporal distribution of dust emission fluxes and transport pathways. Figure 11 illustrates the average dust deposition flux for the two EOF modes and their difference compared to the June-July 2013 mean, which are linked to different regimes of the atmospheric circulation and subsequent the regional distribution of dust source activations and predominant dust transport

routes as presented above. Generally, the dry and the total dust deposition flux (Figs. 11a, b, e, and f) are both strongest over the North African continent near-by the source regions. There, the atmospheric dust load is highest and also the contribution of large particles to the atmospheric dust burden is significantly increased compared to remote regions. Due to gravitational settling, smaller particles remain longer within the atmosphere than larger particles, which contribute strongly to the dust

deposition flux near-by the source regions. Thus, the spatial distribution of dust deposition fluxes over the North African continent is implicitly affected by the spatial distribution of dust emission fluxes and thus reflects the dust source activity as shown in Figs. 7a and b. Wet deposition caused by scavenging processes related to clouds and precipitation are in particular relevant over mountain regions and where fronts and precipitation occur (Figs. 11c, d).

With regard to the EOF phases representing different modes in atmospheric circulation (cf. Fig. 5), dust deposition rates

are increased over the western Mediterranean basin during the negative EOF phase (Fig. 11), which is in agreement with increased northward dust transport as shown in Fig. 9 and Fig. 10 and the generally increased atmospheric dust concentration over the western basin (Fig. 8b). During the positive EOF phase, dust deposition fluxes are generally at a lower level, which is in agreement with lower levels of northward dust transport and atmospheric dust loadings over the Mediterranean. As wet deposition is quite efficient in removing dust from the atmosphere, fronts and associated bands of precipitation result in

significant spatial inhomogeneities in dust deposition rates, which variability is linked to the different modes of atmospheric circulation (Figs. 11c and d). The imprints of wet deposition are even dominating the total deposition flux as shown in Figs. 11e and f.

## 7    Discussion

The atmospheric circulation, in particular the distribution of pressure patterns, is one of the major controls on the atmospheric

dust life-cycle. Pressure gradients determine local wind speed distributions that foster or inhibit dust emission, and dust transport pathways follow the resulting air flow. As the pressure patterns change their geographic location and intensity with time, pressure gradients change and consequently winds change in direction and speed as well. Here, we applied the EOF analysis to identify coherent changes in pressure patterns, represented by the 850 hPa geopotential height. We have chosen the 850 hPa level for this analysis as this level is present over mountains and less affected by local characteristics as the 1000 hPa level.

It further is situated in the convective boundary layer and therefore related to the surface winds. The 850 hPa level marks the lower height for dust transport off the Mediterranean coast of North Africa, however, the EOF analysis of higher levels (e.g., 700 hPa) show similar patterns (not shown). The first PC for the June to July 2013 period reflects the northward propagation of the subtropical high (Azores high) and associated changes in the pressure patterns (Fig. 5), which describes 45% of the variance. Based on the assumption that the activation of major dust source regions and dust transport routes are determined by wind

fields that develop from pressure gradients, the first EOF is assumed to represent dominant variances in the spatial distribution of dust sources activation, dust emission fluxes and transport routes. To compose characteristics representing the opposed phases identified from the EOF analysis (Fig. 4), days are assigned to a negative or a positive phase according to the first PC, which is characterized by a predominant high pressure zone over the western and central Mediterranean for the positive phase,

and a predominant heat low trough extending from the western Grand Erg Occidental toward the Gulf of Biscay for the negative phase, respectively. This classification allows for a separate consideration of dust conditions linked to atmospheric circulation patterns related to reduced AOD levels over the western Mediterranean during positive phases and enhanced levels of dust loading during negative phases. This dipole pattern is also evident from the AERONET AOD estimates, where the Gozo station

(Malta) records enhanced AOD levels at times when the AERONET station Palma de Mallorca (Spain) experiences in many cases low AOD levels. Eventually, the classification via EOF (PC) analysis allows for an investigation of the atmospheric dust life-cycle depending on the mode of atmospheric circulation, in particular the position of the subtropical ridge (western versus central Mediterranean) and the phase of the SHL (eastern or western phase following *Chauvin et al.* (2011)). Given the fact that dust is present almost every day in the atmosphere over North Africa, the level of dust source activation and, consequently,

the strength of dust emission fluxes, and the dust transport direction determine the characteristics of the atmospheric dust cycle regarding its predominance and the distance and direction of dust transport routes. Because nevertheless, dust emission marks the beginning of the atmospheric dust life-cycle and deposition its termination. Relating the predominance of the individual elements of the dust life-cycle (emission - transport - deposition) to prevailing atmospheric circulation pattern referring to pressure distributions, allows for analyzing the net-balance in general and the combination of the elements in particular, which

in concert determine the distribution of the dust concentration in the atmosphere. In relation to the ChArMEx/ADRIMED SOP in June and July 2013, and exemplarily for the month when dust concentrations over the Mediterranean basin are at an increased level (cf. *Nabat et al.*, 2013; *Rea et al.*, 2015), the northward net dust transport at 37° N, which corresponds to the export of dust toward the western Mediterranean basin, is on average about three to four times stronger during negative phases ($480 \, \text{kg} \, \text{day}^{-1}$) than during positive phases ($140 \, \text{kg} \, \text{day}^{-1}$) over the longitudinal range of -5° E to 20° E. With regard to the

June-July average, the northward dust flux is increased during negative phases by 45% (145% of mean value) and decreased by 56% during positive phases (44% of mean value). The predominance of dust source regions and dust transport routes also impact on the dust deposition fluxes, and thus ultimately on the delivery of micro-nutrients to the ecosystems where the dust is deposited. However, dust deposition rates over the Mediterranean basin are additionally determined by atmospheric stability, and cloud and precipitation formation processes.

EOF analysis of long-term reanalysis fields such as from the ERA-Interim product reveal an interannual variability of the predominance of the negative respective positive EOF phase and consequently of the related atmospheric circulation pattern (Fig. 6). Composites from the atmosphere-dust model system COSMO-MUSCAT illustrate a link between phase of the EOF - classifying the related atmospheric circulation - and different elements of the atmospheric dust life-cycle. As dust emission and transport is a direct function of the wind, which is determined by pressure gradients that result from the atmospheric circulation,

this link is also suggested by the physical understanding of the atmospheric dust life-cycle. In the frame of this study, the predominance of atmospheric circulation patterns determining dust export toward the Mediterranean basin and southern Europe is in focus. Hence, the number of days that can be classified as either negative or positive EOF are relevant. The variability in northward dust export due to the atmospheric circulation is elaborated in detail exemplarily for June-July 2013 by choosing the meso-scale model COSMO-MUSCAT, which simulates the atmosphere and the dust life-cycle as simultaneously as possible.

This way the distribution of dust in the atmosphere is consistent with the simulated state of the atmosphere. Although the

EOF analysis from 37-years of ERA-Interim reanalysis fields as presented in section 4 provides first insights into the interannual variability that contribute to the variability in atmospheric dust emission conditions and transport capacities, simulations from climate models with on-line coupled dust modules such as ECHAM6-HAM2 (*Heinold et al.*, 2016) are required to fully investigate the links between the predominance of atmospheric circulation pattern and dust export fluxes.

## 8   Conclusions

This study aimed for an assessment on atmospheric circulation patterns that determine dust source activation and dust transport toward the western Mediterranean basin with regard to the ChArMEx/ADRIMED special observation period in June and July 2013. Simulations using the atmosphere-dust model COSMO-MUSCAT allow for a complementary investigation of the atmospheric dust life-cycle, in particular the spatio-temporal distribution of dust source activations, dust emission fluxes, dust transport pathways and deposition, and their dependence on the atmospheric circulation and their variability.

In order to express the atmospheric variability, EOF analysis was performed and used to identify different modes of variance: The first EOF explains 45% of the variance and identifies therefor the dominant variability in 850 hPa geopotential. It can be linked to two opposing pressure patterns: A positive phase, which is characterized by a high pressure ridge extending from the Atlantic into the Mediterranean Sea with the SHL being in its eastern phase (*Chauvin et al.*, 2011), an enhanced pressure gradient forms, in particular over the eastern part of the Grand Erg Occidental. Consequently, a hot spot for dust emission can be found there. The negative phase is associated with a low pressure trough between Bay of Biscay and western Sahara and a weak high pressure zone over central Mediterranean. Hotspot for dust emission appears over the Libyan desert southeast of the Gulf of Sidra, where the pressure gradient between the central Mediterranean high and the Etesian heat low increases. However, increased level of dust source activation and dust emission flux does not directly convert into increased atmospheric dust burden over the Mediterranean. The increased dust source activity over the eastern Grand Erg Occidental is related to prevailing northerly winds, which limit dust transport toward the Mediterranean coast, although an eastward migration of the high pressure zone enables dust transport in northern direction. Dust emitted from sources located in the Libyan desert southeast of the Gulf of Sidra is predominantly transported in southward direction into the Sahara. Meridional dust transport toward the Mediterranean during June and July 2013 predominantly occurred between 5° W and 10° E when the SHL was in its western phase and the high pressure ridge was centered over the central Mediterranean.

In conclusion, the EOF analysis of the 850 hPa geopotential height is valuable for the identification of general atmospheric circulation pattern. Here, patterns of coherent variance are linked to pressure patterns that foster dust source activation over particular regions. Due to the dependence of dust transport on wind direction, dust export toward the Mediterranean basin and consequently atmospheric dust concentrations over selected cities (AERONET sites), and dust deposition rates can be linked to that particular atmospheric circulation pattern as well. In terms of dust optical and physiochemical properties, which are determined by their source region, certain pressure pattern enable for transport of dust predominantly from particular dust source regions. As also dust transport routes change with the location of pressure centers, internal and external mixing of dust plumes with further aerosols and pollutants is expected. As dust delivers micro-nutrients to the eco-systems, dust deposition

fluxes may impact on the bio-productivity of individual ecosystems. Long-term simulations such as reanalysis data sets or the CMIP (Coupled Model Intercomparison Project) data sets motivates studies examining the trend of the occurrence of atmospheric circulation patterns associated with the negative or positive phase - and hence the predominance of dust transport toward the Mediterranean basin. As several densely populated regions are situated in coastal Mediterranean region, the occurrence frequency of dust haze is relevant for e.g. air quality and thus contributes to quality of life and well being of about 150 million people living along the shorelines of the Mediterranean Sea (United Nations Environment Programme (UNEP), http://www.unepmap.org). Furthermore, atmospheric dust alters the radiation budget, impacts on cloud and precipitation formation processes and affects the ecosystems, which makes the role of atmospheric dust vital for the Mediterranean climate, its environment and anthroposphere.

*Acknowledgements.* KS and MU acknowledge funding through the Leibniz Association for the Project "Dust at the Interface - modelling and remote sensing". The authors thank Ina Tegen for fruitful discussion, the Deutscher Wetterdienst for support and cooperation, and the AERONET team from the stations at Gozo (PI: Ray Ellul), Murcia (PI: Juan Ramon Moreta Gonzalez), Oujda (PI: Diouri Mohammed and Djamaleddine Chabane), Palma de Mallorca (PI: Juan Ramon Moreta Gonzalez), and Tizi Ouzou (PI: Zohra Ameur) for obtaining the measurements and providing the level 2.0 coarse mode AOD data. The authors further thank NASA GES DISC for development and maintenance of Giovanni online system providing MODIS collection 6 aerosol products and MERRA dust AOD fields. ERA-Interim data were made available through the ECWMF. This study contributes to ChArMEx WP3 on transport processes.

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

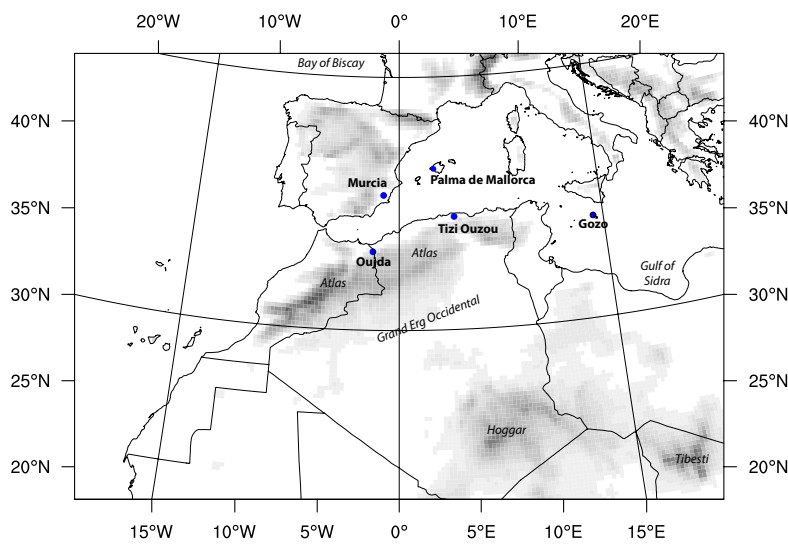

**Figure 1.** Overview on AERONET measurement sites considered within this study: Gozo, Malta (36.03° N; 14.26° E), Murcia, Spain (38.00° N; -1.17° E), Oujda, Morocco (34.65° N; -1.90° E), Palma de Mallorca, Spain (39.55° N; 2.63° E), Tizi Ouzou, Algeria (36.70° N; 4.06° E). Orography is indicated by gray shading.

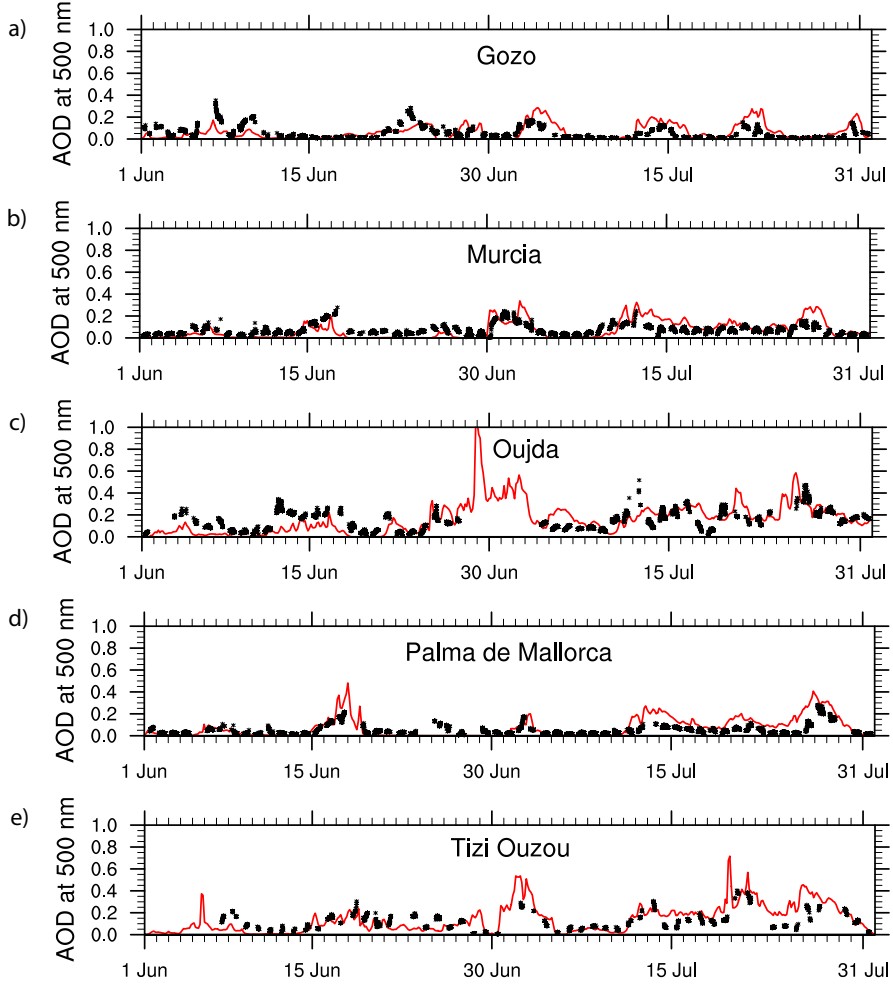

**Figure 2.** Temporal evolution of dust AOD at five different stations (see Fig. 1) located along frequent dust transport routes. Dust AODs calculated from 3-hourly COSMO-MUSCAT output for June and July 2013 are represented by the red line, AERONET level 2.0 coarse mode AOD retrievals from sun-photometer measurements are shown in black. COSMO-MUSCAT dust AODs are calculated for the grid cell corresponding to the AERONET measurement site.

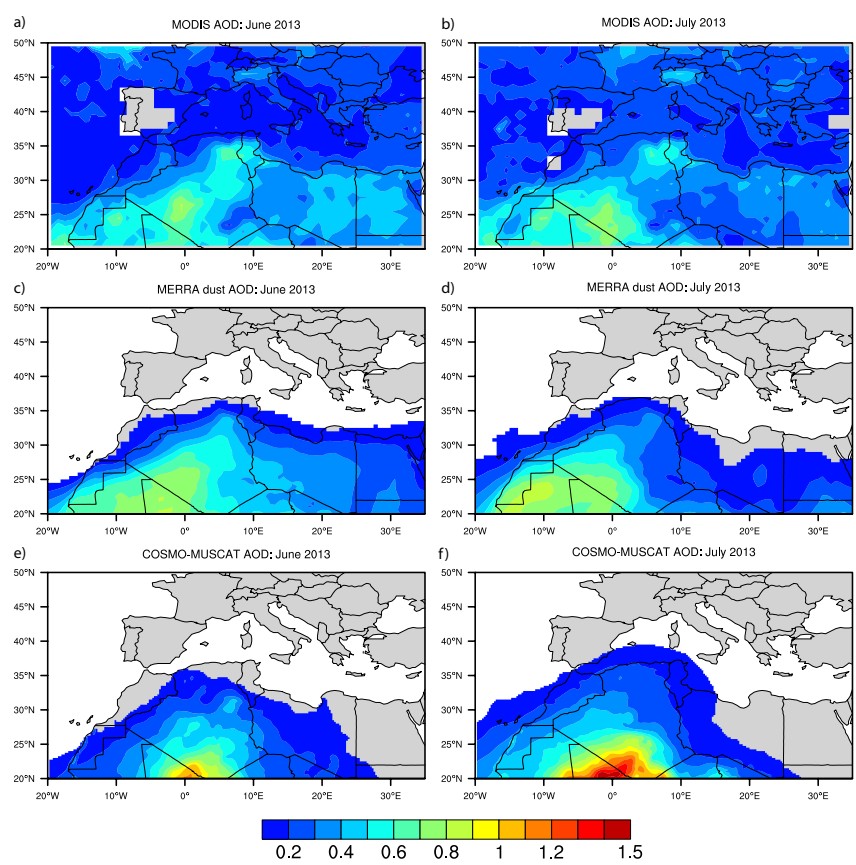

**Figure 3.** Spatial distribution of dust AODs over North Africa and the Mediterranean basin shown as monthly mean values. Upper panel shows the monthly mean AOD distribution as estimated from Terra and Aqua MODIS observations (MODIS Collection 6 merged aerosol product) for June (a) and July (b) 2013. Middle panel shows June (c) and July (d) mean dust AOD taken from the MERRA dataset for 2013. Bottom panel shows monthly mean dust AOD distribution as estimated from COSMO-MUSCAT 12 UTC simulations for June (e) and July (f) 2013.

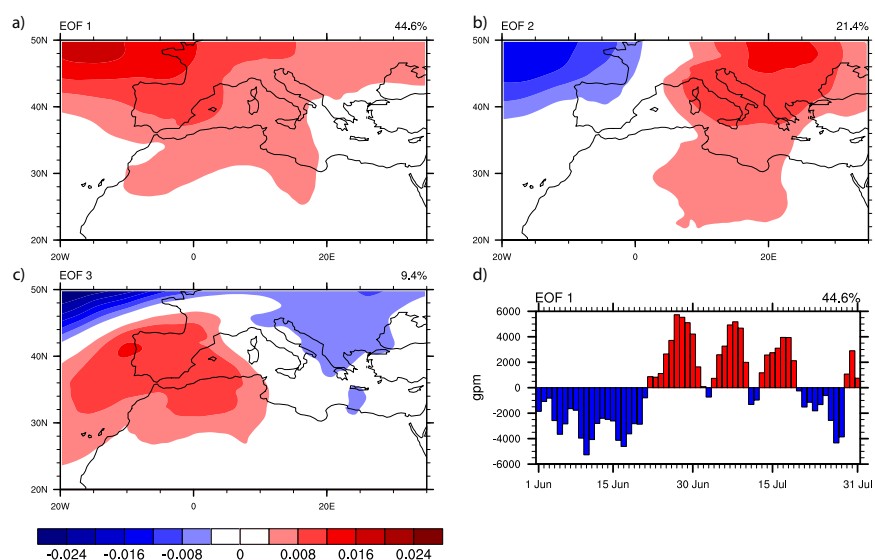

**Figure 4.** EOF analysis of the 00 UTC geopotential height at 850 hPa taken from COSMO-MUSCAT simulations for June and July 2013. Shown are the first three EOFs and the principal component of the first EOF. The variance explained is given in percentage (upper right corner of each plot). Negative phase: 1 - 21 June, 3 July, 11 - 12 July, 20 - 28 July 2013. Positive phase: 22 June - 2 July, 4 - 10 July, 13 - 19 July, 29 - 31 July 2013.

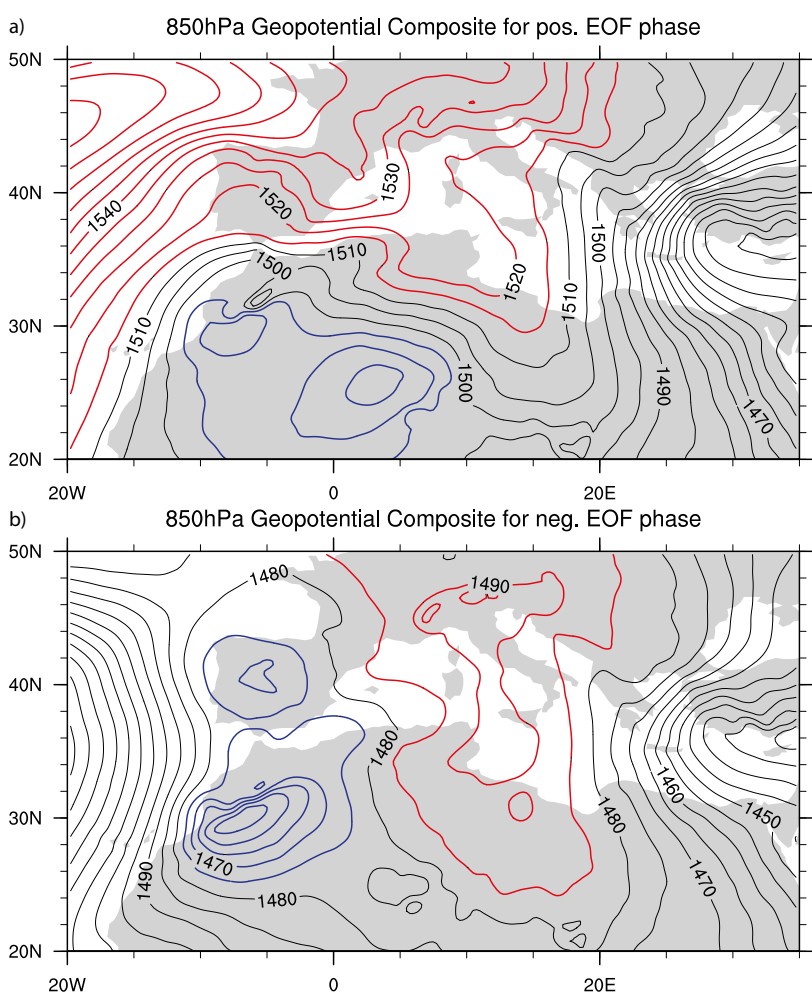

**Figure 5.** Composite of the 850 hPa geopotential height [gpm] for positive (a) and negative (b) phases of the first EOF for June to July 2013.

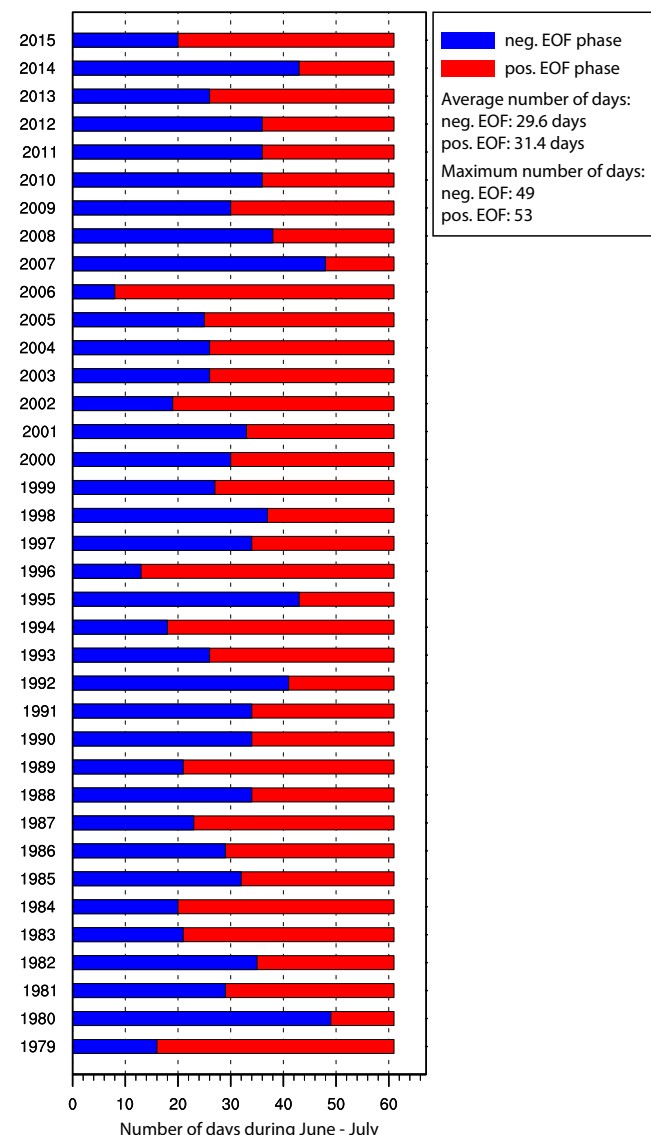

**Figure 6.** Statistic on number of days identified as negative respective as positive EOF phase during June-July period for each year during 1979-2015. EOF is calculated from 00 UTC ERA-Interim 850 hPa geopotential fields.

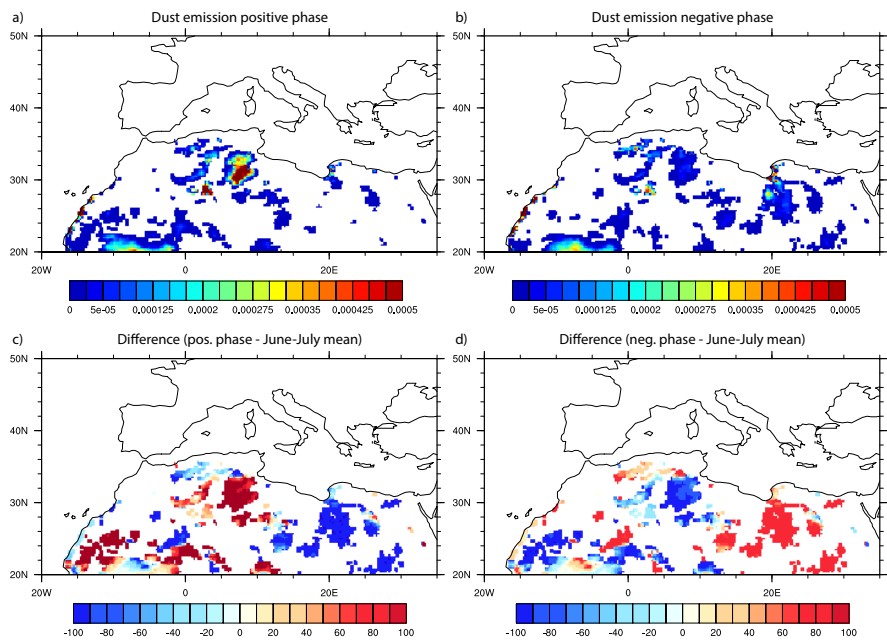

**Figure 7.** Composite of the COSMO-MUSCAT model dust emission flux [kg m$^{-2}$ day$^{-1}$] for positive (a) and negative (b) phases of the first EOF for June to July 2013. (c) and (d) show the difference [%] of the average dust emission flux during (a) and (b) compared to the June-July 2013 mean.

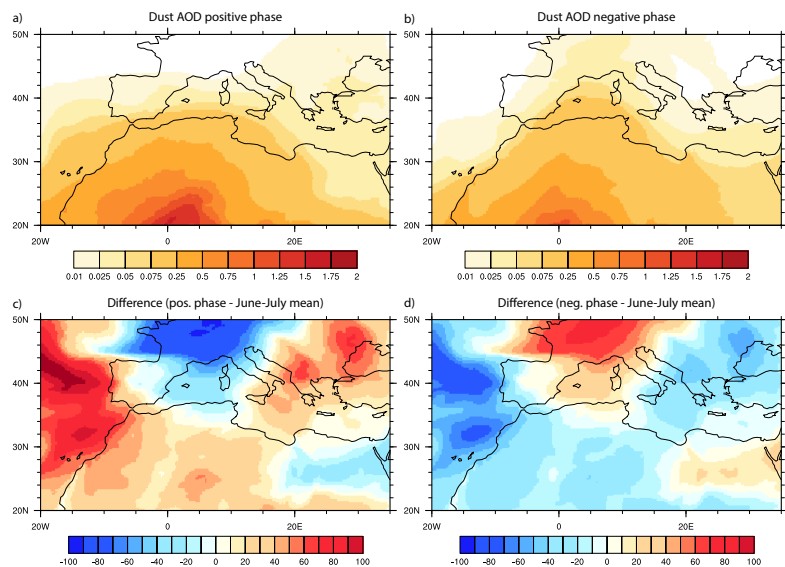

**Figure 8.** Composite of the dust AOD for positive (a) and negative (b) phases of the first EOF for June to July 2013. (c) and (d) show the difference [%] regarding the June-July 2013 mean.

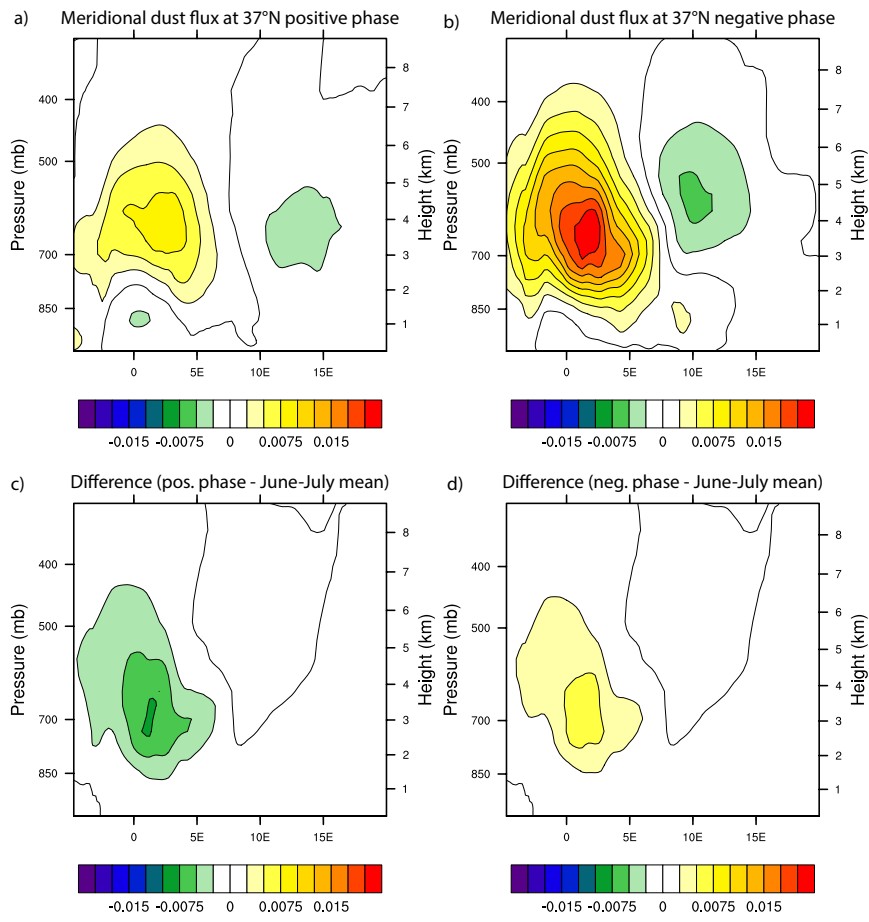

**Figure 9.** Composite of the meridional dust flux at $37°$ N [kg m$^{-2}$ day$^{-1}$] for positive (a) and negative (b) phases of the first EOF for June to July 2013. (c) and (d) show the difference [kg m$^{-2}$ day$^{-1}$] regarding the June-July 2013 mean.

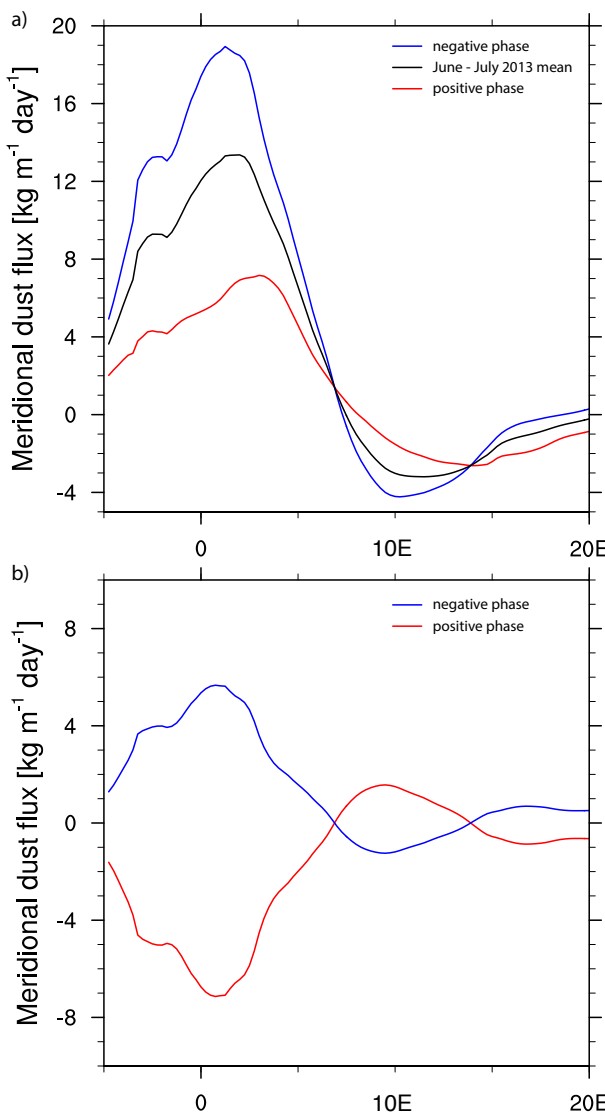

**Figure 10.** (a) Composite of the vertical totals of the meridional dust flux at $37^\circ$ N [kg m$^{-1}$ day$^{-1}$] for positive (red) and negative (blue) phases of the first EOF for June to July 2013 as well as the June-July 2013 mean flux. (b) shows the difference [kg m$^{-1}$ day$^{-1}$] regarding the June-July 2013 mean.

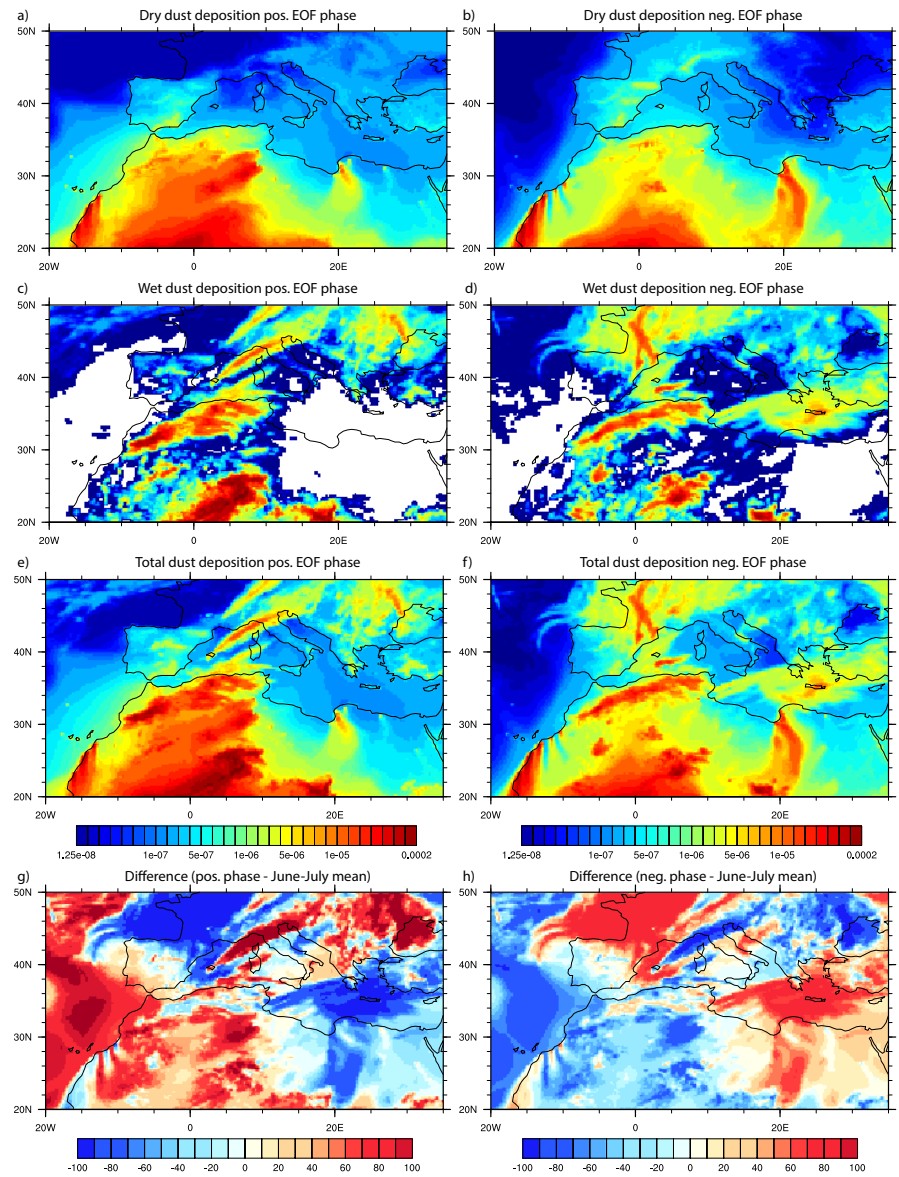

**Figure 11.** Composite of the dry dust deposition [kg m$^{-2}$ day$^{-1}$] for positive (a) and negative (b) phases of the first EOF for June to July 2013. Similar composites for wet deposition and total (dry + wet) deposition fluxes are shown in (c) and (d) respectively in (e) and (f). (g) and (h) show the difference [%] of the average total dust deposition flux during (e) and (f) compared to the June-July 2013 mean.