# Peer review of "North African dust transport toward the western Mediterranean basin: Atmospheric controls on dust source activation and transport pathways during June-July 2013"

_Atmospheric Chemistry and Physics, 2016_

## Referee Comment (RC1) · Anonymous Referee #1 · 31 Jul 2016

Review of: 'North African dust transport toward the western Mediterranean basin: Atmospheric controls on dust source activation and transport pathways during June-July 2013' by K.Schepanski et al.

This article presents an analysis of the tropospheric mineral dust transport from Africa to Western Europe and during the Charmex field campaign of summer 2013. The topic is very important, mineral dust being difficult to measure and to model, but having a large impact on particulate matter concentrations in the troposphere. The use of EOF is an original way to sort multiple and complex meteorological events, combined to

complex soil and surface properties, leading to huge difficulties to know where and when mineral dust emissions may occur. But several questions remains, which are listed below in the 'Major remarks'. Some other minor remarks are also proposed at the end of this review.

This article may be accepted but after major revisions.

Major remarks: The main concern is the lack of originality of the paperĂă: the use of a statistical analysis is original but the main goal of the paper is not. A lot of papers are already published about this kind of transport and over the Mediterranean. These articles are abundant in the literature (mainly ACP and JGR-atm), including the ACP/AMT Charmex special section. We recommend to the authors to better reference the recent studies and to extract a new way to introduce the results in order to be really original. A suggestion: estimate EOF for severals years over the region (using GFS or ECMWF model outputs for example) and characterize the specific year of 2013 in this ensemble.Then, using the already modeled period, conclude if 2013 led to less/more mineral dust from Africa to Europe. This suggestion requires to extend and improve the EOF prt of this paper. But I think this could give a real originality of the used approach and to this study.

The second lack is the validation of the modeled meteorology and the mineral dust production model used. It is clear that mineral dust emissions are a combination of 'favorable' meteorology (surface wind speed) and 'favorable' soils and surface (including roughness length, soil humidity, vegetation, topography). The accuracy of the result will be the multiplication of these two large uncertainties. But the modeled wind speed is not validated and the mineral dust production model is an old one, with a large set of uncertainties: the vertical dust flux is tabulated with a constant, the number of bins are low. In addition, the model is regional and applied to a period corresponding to an intensive field campaign: numerous papers are on the ACP/AMT section. Why are they not usedĂă?

The third lack is the differences between the used tools and the goal of the paper. The use of EOF and meteorology at 850hPa is a climatological approach. This provides informations on long-range transport only (and certainly not on surface wind speed, the main engine for mineral dust production). On the other hand, the simulation is carried on for two months only: perhaps a specific case, not representative of general circulations, this has to be evaluated and discussed. This remark may be smoothed by extending the paper as suggested in remark #1.

Minor remarks:

- The abstract is too long. New results must be better highlighted.

This sentence shows the confusion about the meteorological scales in the studyÂǎ: 'The study elaborates the question on the variability of dust transport toward the Mediterranean and Europe in dependence on the atmospheric circulation as a driver for dust emission and a determinant for dust transport routes...'. The atmospheric circulation is not the driver of emissions. It is only a driver for transport, once dust are emitted.

- Introduction: A key point is the well-cited publication of (Shao, 2011). This shows this kind of study was already done. Perhaps the authors may extend the presentation of this publication to better place their own findings. Same remark for the studies of (Moulin et al.): the influence of NAO was deeply studied in these papers and their results could be better presented.

2 Data and methods:

- Definition of wind shear stress could be deleted, being well known. For the model introduction, please add more details on the uncertainties.

- p5.l.5: The alpha constant is not defined. But this could clearly be a very important parameter.

- p5.l.20: If the model is on-line, the shape of the dust (and the related constants) may

have an effect on AOD but also on direct and indirect effects. Please clarify.

2.2 Validation of simulations using only AOD is frequent for global models. But may appear too simple for regional models. The paper could be improve using more and finest data, especially because the study is linked to an intensive filed campaign.

2.3 Indeed the EOF are designed for long-time period. Please discuss the fact this tool is only used for a short period. What is the representativity of the results in this case. Or think to the suggestion #1 of this review.

3.

This section is interesting, a good bibliography but very long and a mixture of several topics. The first part is close to the introduction (some references are the same) and the second part presents applications of EOF: the topic of section 2. Please simplify and merge these three sections. Results for EOF could be in a new section 3: 'Meteorological validation against measurements and EOF results'.

4.

p.9, l.9: 'Dust source activation...' the concept for dust emissions (meteorology and soil/surface) was already described and cited several times before. The authors may be more synthetic and directly goes to the new results.

5.4 Dust deposition. This is an interesting section, but a validation to existing data is necessary before to conclude with the model only. In particular, the wet scavenging is often roughly designed in the models and the uncertainty is important.

ConclusionÂă: The end of the conclusion is more related to a bibliography. Please focus on your results only.

———————————————————

---

## Referee Comment (RC2) · Anonymous Referee #2 · 8 Aug 2016

General remarks:

The present manuscript search to aim for an assessment on atmospheric circulation patterns that determine dust source activation and dust transport toward the western Mediterranean basin with regard to the ChArMEx/ADRIMED special observation period in June and July 2013. EOF analysis is used to identify different modes of variance of dust simulations using the atmosphere-dust model COSMO-MUSCAT.

While the results of the study are interesting to be published, their presentation and discussion are not yet sufficient to be published at Atmospheric Chemistry and Physics

in the current form. The present manuscript is focusing on the meteorological synoptic patterns that determine the dust transport towards the western Mediterranean basin by means a regional dust model, the COSMO-MUSCAT. The main problem is that the authors do not clearly provide evidence of the performance of the meteorological-dust model for the study period to provide support for the conclusions. ChArMEx project proposes a multi-scale model-observation integrated strategy with satellite and field observations. During the Charmex short observation periods (SOP) detailed process studies are performed during intensive campaigns (for summer 2012 and 2013); studies include continental plume transport and aging and chemical and optical closures in the column. In this sense, I would suggest a comparison with satellite aerosol products observations (as MODIS or SEVIRI/AERUS) or ground-based lidar stations (such as EARLINET) in addition to reanalysis (such as MACC reanalysis and ERA-Interim) to demonstrate the ability of the COSMO-MUSCAT model to reproduce the dust transport during summer 2013 over the Mediterranean.

Finally, I would suggest to include in the discussion of the results some recent references as the following:

- Cuevas, E., Gómez-Peláez, Á. J., Rodríguez, S., Terradellas, E., Basart, S., García, R. D., García, O. E., and Alonso-Pérez, S.: Pivotal role of the North African Dipole Intensity (NAFDI) on alternate Saharan dust export over the North Atlantic and the Mediterranean, and relationship with the Saharan Heat Low and mid-latitude Rossby waves, Atmos. Chem. Phys. Discuss., doi:10.5194/acp-2016-287, 2016. In this manuscript it is revised the index that quantifies the North African Dipole Intensity (NAFDI), and explain its relationship with the Saharan Heat Low (SHL) and mid-latitude Rossby waves. If you check the results of this work, you will see similar results to your negative phase associated of the dust transport and meteorological patterns associated with the negative NAFDI phase.

- Menut, L., Rea, G., Mailler, S., Khvorostyanov, D., and Turquety, S.: Aerosol forecast over the Mediterranean area during July 2013 (ADRIMED/CHARMEX), Atmos.

Chem. Phys., 15, 7897-7911, doi:10.5194/acp-15-7897-2015, 2015. This work is also covering the same study period using the CHIMERE model and it would desirable to compare your results with those included in this analysis.

Minor errors:

Page 2 Line 28. You could include the following reference: Gkikas, A., Basart, S., Hatzianastassiou, N., Marinou, E., Amiridis, V., Kazadzis, S., Pey, J., Querol, X., Jorba, O., Gassó, S., and Baldasano, J. M.: Mediterranean intense desert dust outbreaks and their vertical structure based on remote sensing data, Atmos. Chem. Phys., 16, 8609-8642, doi:10.5194/acp-16-8609-2016, 2016.

Page 2 Line 33. If you check Ginoux et al. (2012) you will see that there are some desert dust anthropogenic sources that affects the Mediterranean.

- Ginoux, P., Prospero, J. M., Gill, T. E., Hsu, N. C., & Zhao, M. (2012). Global‐scale attribution of anthropogenic and natural dust sources and their emission rates based on MODIS Deep Blue aerosol products. Reviews of Geophysics, 50(3).

Page 3 Line 14. You could add Gkikas et al. (2016).

Page 3 Line 26. Could you give any detail about the radiative module implemented in the model?

Page 4 Line 32. Does the model include any soil moisture or drag partition correction in the calculation of the threshold friction velocity?

Page 5 Line 26. You could add Gkikas et al. (2016).

Page 6 Line 8. Could you include any other information about the model configuration used in the present study? As the meteorological initial and boundary conditions, and the dust initial conditions (does the model include data assimilation?)

Page 6 Line 17. For the coarse-mode AOD comparison with AERONET, are you are taking the coarse fraction of the simulated dust fields (i.e. r > 1microm)?

Page 7 Line 16. You could add the work of Rodríguez, S., Querol, X., Alastuey, A., Kallos, G., and Kakaliagou, O.: Saharan dust contributions to PM10 and TSP levels in Southern and Eastern Spain, Atmos. Environ., 35, 29 2433–2447, 2001.

Page 8 Line 2. You could add the work of Basart, S., Pérez, C., Cuevas, E., Baldasano, J. M., and Gobbi, G. P.: Aerosol characterization in Northern Africa, Northeastern Atlantic, Mediterranean Basin and Middle East from direct-sun AERONET observations, Atmos. Chem. Phys., 9, 8265-8282, doi:10.5194/acp-9-8265-2009, 2009. Page 8 Line 10. For the Eastern Mediterranean region, you should consider the Middle East desert dust sources particularly in spring.

Page 8 Line 16. Could you quantify "significantly stronger"?

Page 9 Line 13. You should indicated that they are the dust sources predicted by the model.

Section 5.1. In addition to AERONET comparison, you should include the spatial comparison of your model results using other aerosol observational datasets as MODIS, SEVIRI/AERUS and EARLINET.

Figure 1. Include the locations of Sirtra and Biscaia Gulf.

---

## Editor Comment (EC1) · F. Dulac (Editor) · 31 Aug 2016

I find that the present focus on summer 2013 is a strong limitiation of the manuscript. One possible interesting and somewhat expected outcome of such a study would be a discussion about the 2013 campaign period representativeness based on the analysis of a larger data set including several summers. I would therefore recommend that this point is considered during the likely revision of the paper.

---

## Author Comment (AC1) · 17 Oct 2016

**Point-by-point reply to the reviewer's comments**

*The authors would like to thank the reviewers for the time they spend on the manuscript, and for providing helpful and constructive comments and suggestions. We have considered carefully all comments made; please find our detailed reply (italic) below.*

**Review #1**

This article presents an analysis of the tropospheric mineral dust transport from Africa to Western Europe and during the Charmex field campaign of summer 2013. The topic is very important, mineral dust being difficult to measure and to model, but having a large impact on particulate matter concentrations in the troposphere. The use of EOF is an original way to sort multiple and complex meteorological events, combined to complex soil and surface properties, leading to huge difficulties to know where and when mineral dust emissions may occur. But several questions remains, which are listed below in the 'Major remarks'. Some other minor remarks are also proposed at the end of this review. This article may be accepted but after major revisions.

*Many thanks for your time spend on the manuscript, your encouraging assessment and helpful comments. Please find our detailed reply below.*

Major remarks:
The main concern is the lack of originality of the paper: the use of a statistical analysis is original but the main goal of the paper is not. A lot of papers are already published about this kind of transport and over the Mediterranean. These articles are abundant in the literature (mainly ACP and JGR-atm), including the ACP/AMT Charmex special section. We recommend to the authors to better reference the recent studies and to extract a new way to introduce the results in order to be really original. A suggestion: estimate EOF for severals years over the region (using GFS or ECMWF model outputs for example) and characterize the specific year of 2013 in this ensemble. Then, using the already modeled period, conclude if 2013 led to less/more mineral dust from Africa to Europe. This suggestion requires to extend and improve the EOF part of this paper. But I think this could give a real originality of the used approach and to this study.

*Many thanks for this comment! We have followed the reviewers (and editors) suggestion and extended the EOF part of this manuscript. We have calculated the EOF from ERA-Interim geopotential fields for 1979-2015. Comparing the EOF calculated from this 37-year period to those calculated from the 2013, the same patterns occur as shown on the figure below. To assure comparability between the analysis of the meso-scale model COSMO-MUSCAT and the global-scale ERA-Interim reanalysis, the EOF analysis is done for the June-July 2013 period as well as for the June-July 1979-2015 period.*

[revised manuscript text omitted]

The second lack is the validation of the modeled meteorology and the mineral dust production model used. It is clear that mineral dust emissions are a combination of 'favorable' meteorology (surface wind speed) and 'favorable' soils and surface (including roughness length, soil humidity, vegetation, topography). The accuracy of the result will be the multiplication of these two large uncertainties. But the modeled wind speed is not validated and the mineral dust production model is an old one, with a large set of uncertainties: the vertical dust flux is tabulated with a constant, the number of bins are low. In addition, the model is regional and applied to a period corresponding to an intensive field campaign: numerous papers are on the ACP/AMT section. Why are they not used?

*Regarding the general model accuracy and uncertainty, the model has been extensively tested in the past with observations from several field studies and available station observations and remote sensing data (Heinold et al., 2009, 2011, Schepanski et al., 2009, 2015, Tegen et al., 2013, Niedermeier et al., 2014). In the frame of ChArMEx, COSMO-MUSCAT model simulations were validated in contribution to two publications: Mallet et al. (2016) and Granados et al. (2016). As the present study aims for elaborating the variability of dust export toward the Mediterranean, which somehow affects the examination of results from the ChArMEx project, we decided to not examine individual case studies for the sake of a clear manuscript agenda.*

The third lack is the differences between the used tools and the goal of the paper. The use of EOF and meteorology at 850hPa is a climatological approach. This provides informations on long-range transport only (and certainly not on surface wind speed, the main engine for mineral dust production). On the other hand, the simulation is carried on for two months only: perhaps a specific case, not representative of general circulations, this has to be evaluated and discussed. This remark may be smoothed by extending the paper as suggested in remark #1.

*Please see our reply above. We have extended the "climatological approach" and discuss the summer 2013 (June-July) with regard to the summers 1979-2015.*

Minor remarks:
- The abstract is too long. New results must be better highlighted. This sentence shows the confusion about the meteorological scales in the study: 'The study elaborates the question on the variability of dust transport toward the Mediterranean and Europe in dependence on the atmospheric circulation as a driver for dust emission and a determinant for dust transport routes...'. The atmospheric circulation is not the driver of emissions. It is only a driver for transport, once dust are emitted.

*Many thanks for your comment. We have shortened and revised the abstract.*
*We agree, dust emission is determined (and limited) by both, soil conditions and atmospheric conditions. In meteorological terminology, however, 'atmospheric circulation' (see e.g. AMS glossary) refers to the large-scale synoptic features and their interplay, i.e., pressure and wind systems. Therefore, the atmospheric circulation does drive the (surface) winds that can mobilize and transport mineral dust. In order to clarify, we restate the sentence as follows:*
*"'The study elaborates the question on the variability of dust transport toward the Mediterranean and Europe with regard to the atmospheric circulation conditions controlling emission and transport routes of Saharan dust […]"*

- Introduction: A key point is the well-cited publication of (Shao, 2011). This shows this kind of study

was already done. Perhaps the authors may extend the presentation of this publication to better place their own findings. Same remark for the studies of (Moulin et al.): the influence of NAO was deeply studied in these papers and their results could be better presented.

*Many thanks for your comment; we have added a paragraph describing the results of Moulin et al., in particular the influence of the NAO on dust emission over North Africa:*

*"Moulin et al. (1997) propose a link between the spatial distribution and the phase of the North Atlantic Oscillation, which is described by an index reflecting the pressure difference between Icelandic low and Azores high. The authors conclude, that the seasonal variations in pressure difference over the North Atlantic, in particular the modulation of the atmospheric circulation over the North Atlantic - European sector, impacts on the North African atmospheric dust life-cycle. Consequently, a high positive NAO index, characterized by a deepening of the Icelandic low and a strong Azores high, fosters drier conditions over North Africa and thus enhances the chances for dust mobilization."*

2 Data and methods:
- Definition of wind shear stress could be deleted, being well known. For the model introduction, please add more details on the uncertainties.

*The definition of the wind shear stress is removed. For model uncertainties we refer to previous studies using the dust model version of COSMO-MUSCAT. The model has been extensively tested with observations from several field studies and available station and remote sensing data (Heinold et al., 2009, 2011; Schepanski et al., 2009; Tegen et al., 2013; Niedermeier et al., 2014).*

- p5.l.5: The alpha constant is not defined. But this could clearly be a very important parameter.

*The sandblasting efficiency alpha was introduced earlier (page 4, line 21), however, it is defined here again.*

- p5.l.20: If the model is on-line, the shape of the dust (and the related constants) may have an effect on AOD but also on direct and indirect effects. Please clarify.

*Dust-radiation interactions are computed online at solar and thermal wavelength bands and account for variations in the simulated size-bin resolved aerosol concentrations (Helmert et al., 2007). It can impact on the meteorology and consequently implicitly feed back on dust emission and dust transport (Heinold et al., 2008). As we already describe in detail on page 5, the dust optical properties are computed using Mie theory, which requires assuming spherically shaped particles. Although this assumption usually does not hold for mineral dust, the errors in radiative flux computation are small when integrating over hemispheres (Mishchenko et al., 1995; Seinfeld and Pandis, 1998). Otto et al. (2009) showed that AOD, single scattering albedo, and asymmetry parameter were in error by 3.5%, 1%, and 4% respectively if spherical instead of non-spherical particles were assumed. Based on this, the shape effect on dust AOD is negligible relative to the other uncertainties in an atmospheric model.*

2.2 Validation of simulations using only AOD is frequent for global models. But may appear too simple for regional models. The paper could be improve using more and finest data, especially because the study is linked to an intensive filed campaign.

*We do not see what is wrong about using AOD sun photometer measurements, which are very robust and accurate. Together with in-situ concentration measurements they belong to the highest quality data available. However, ground based in-situ observations may not be representative for dust transport within the atmospheric column. For a better spatial evaluation of our model results, we have added a comparison with MODIS collection 6 AOD products.*

2.3 Indeed the EOF are designed for long-time period. Please discuss the fact this tool is only used for

a short period. What is the representativity of the results in this case. Or think to the suggestion #1 of this review.

*We have seized your suggestion #1 and brought the results from the EOF analysis for the June-July 2013 period in the context of a 37-year period (1979-2015).*

3. This section is interesting, a good bibliography but very long and a mixture of several topics. The first part is close to the introduction (some references are the same) and the second part presents applications of EOF: the topic of section 2. Please simplify and merge these three sections. Results for EOF could be in a new section 3: 'Meteorological validation against measurements and EOF results'.

*We followed the reviewer's suggestion and have thematically reordered the bespoke sections. The overview on dust transport pathways (formerly first part of section 3) now follows the Introduction. The EOF analysis builds a new section on its own, section 4.*

4.
p.9, l.9: 'Dust source activation...' the concept for dust emissions (meteorology and soil/surface) was already described and cited several times before. The authors may be more synthetic and directly goes to the new results.

*Many thanks for spotting this! We have cleaned the respective paragraph and results are presented more directly.*

5.4 Dust deposition. This is an interesting section, but a validation to existing data is necessary before to conclude with the model only. In particular, the wet scavenging is often roughly designed in the models and the uncertainty is important.

*We agree that large uncertainties in modelling mineral dust are related to the representation of dry and wet deposition processes. Niedermeier et al. (2014) showed that COSMO-MUSCAT in general does a good job in terms of sedimentation and dry deposition. Wet deposition is difficult to measure, and to our knowledge no data are available here.*

Conclusion: The end of the conclusion is more related to a bibliography. Please focus on your results only.
*We have revised the conclusion section and focus on our results only.*

---

## Author Comment (AC2) · 17 Oct 2016

**Point-by-point reply to the reviewer's comments**

*The authors would like to thank the reviewers for the time they spend on the manuscript, and for providing helpful and constructive comments and suggestions. We have considered carefully all comments made; please find our detailed reply (italic) below.*

**Review #2**

General remarks:
The present manuscript search to aim for an assessment on atmospheric circulation patterns that determine dust source activation and dust transport toward the western Mediterranean basin with regard to the ChArMEx/ADRIMED special observation period in June and July 2013. EOF analysis is used to identify different modes of variance of dust simulations using the atmosphere-dust model COSMO-MUSCAT. While the results of the study are interesting to be published, their presentation and discussion are not yet sufficient to be published at Atmospheric Chemistry and Physics in the current form. The present manuscript is focusing on the meteorological synoptic patterns that determine the dust transport towards the western Mediterranean basin by means a regional dust model, the COSMO-MUSCAT. The main problem is that the authors do not clearly provide evidence of the performance of the meteorological-dust model for the study period to provide support for the conclusions. ChArMEx project proposes a multi-scale model-observation integrated strategy with satellite and field observations. During the Charmex short observation periods (SOP) detailed process studies are performed during intensive campaigns (for summer 2012 and 2013); studies include continental plume transport and aging and chemical and optical closures in the column. In this sense, I would suggest a comparison with satellite aerosol products observations (as MODIS or SEVIRI/AERUS) or ground-based lidar stations (such as EARLINET) in addition to reanalysis (such as MACC reanalysis and ERA-Interim) to demonstrate the ability of the COSMO-MUSCAT model to reproduce the dust transport during summer 2013 over the Mediterranean.

*Many thanks for the time you spend on the manuscript and for your assessment. We have acted on your suggestions and have included a comparison to MODIS aerosol products. Furthermore, the revised version of the manuscript also includes a discussion of the June-July 2013 period with regard to a longer time period using ERA-Interim reanalysis data.*

Finally, I would suggest to include in the discussion of the results some recent references as the following:

- Cuevas, E., Gómez-Peláez, Á. J., Rodríguez, S., Terradellas, E., Basart, S., García, R. D., García, O. E., and Alonso-Pérez, S.: Pivotal role of the North African Dipole Intensity (NAFDI) on alternate Saharan dust export over the North Atlantic and the Mediterranean, and relationship with the Saharan Heat Low and mid-latitude Rossby waves, Atmos. Chem. Phys. Discuss., doi:10.5194/acp-2016-287, 2016. In this manuscript it is revised the index that quantifies the North African Dipole Intensity (NAFDI), and explain its relationship with the Saharan Heat Low (SHL) and mid-latitude Rossby waves. If you check the results of this work, you will see similar results to your negative phase associated of the dust transport and meteorological patterns associated with the negative NAFDI phase.

*Many thanks for your suggestion. As the paper is still under review and following the discussion some changes to the final paper can be expected during the review process, we decided to not include the manuscript at this stage.*

- Menut, L., Rea, G., Mailler, S., Khvorostyanov, D., and Turquety, S.: Aerosol forecast over the Mediterranean area during July 2013 (ADRIMED/CHARMEX), Atmos. Chem. Phys., 15, 7897-7911, doi:10.5194/acp-15-7897-2015, 2015. This work is also covering the same study period using the CHIMERE model and it would desirable to compare your results with those included in this analysis.

*Many thanks for your suggestions. We now refer to this work in the introduction section.*

Minor errors:

Page 2 Line 28. You could include the following reference: Gkikas, A., Basart, S., Hatzianastassiou, N., Marinou, E., Amiridis, V., Kazadzis, S., Pey, J., Querol, X., Jorba, O., Gassó, S., and Baldasano, J. M.: Mediterranean intense desert dust outbreaks and their vertical structure based on remote sensing data, Atmos. Chem. Phys., 16, 8609-8642, doi:10.5194/acp-16-8609-2016, 2016.

> *Many thanks for the suggestion. We now refer to the reference at a later point in the manuscript (see suggestions below).*

Page 2 Line 33. If you check Ginoux et al. (2012) you will see that there are some desert dust anthropogenic sources that affects the Mediterranean.
- Ginoux, P., Prospero, J. M., Gill, T. E., Hsu, N. C., & Zhao, M. (2012). Global scale attribution of anthropogenic and natural dust sources and their emission rates based on MODIS Deep Blue aerosol products. Reviews of Geophysics, 50(3).

> *This may be a misunderstanding here. We do not say that dust from anthropogenic dust sources is not affecting the Mediterranean. The sentence rather means that dust emitted from human-induced source can be controlled by legislative regulations, whereas dust from natural sources, in particular from North African deserts, is rather controlled by weather.*

Page 3 Line 14. You could add Gkikas et al. (2016).

> *Many thanks for the suggestion.*

Page 3 Line 26. Could you give any detail about the radiative module implemented in the model?

> *The radiation scheme developed by Ritter and Geleyn (1992) is implemented in COSMO. Dust-radiation interactions are computed online at solar and thermal wavelength bands and account for variations in the simulated size-bin resolved aerosol concentrations (Helmert et al., 2007). It can impact on the meteorology and consequently implicitly feed back on dust emission and dust transport. This information is added to the manuscript.*

Page 4 Line 32. Does the model include any soil moisture or drag partition correction in the calculation of the threshold friction velocity?

> *The influence of soil moisture on the dust emission flux is considered following Tegen et al. (2002) (cf. their Equation 3). Formally, drag partition in our dust module follows Marticorena & Bergametti (1995).*

Page 5 Line 26. You could add Gkikas et al. (2016).

> *Many thanks for the suggestion. We have added the publication to the list of references.*

Page 6 Line 8. Could you include any other information about the model configuration used in the present study? As the meteorological initial and boundary conditions, and the dust initial conditions (does the model include data assimilation?)

> *Many thanks for spotting this lack of information! We have added the following information to the manuscript: Only dust sources located in North Africa are considered here. Initial and lateral boundary fields are provided by the Deutscher Wetterdienst (DWD, German weather service) global model GME at six-hourly resolution. To keep the meteorology close to the analysis fields, model runs are re-initialized every 48 hours. Following a 24-hour spin-up for the COMSO model, MUSCAT is coupled to COSMO and aerosol processes are computed. Dust concentration fields from the previous cycle are used to initialize atmospheric dust loading in the following cycle. No initialization of the atmospheric dust loading takes place for the first cycle.*

Page 6 Line 17. For the coarse-mode AOD comparison with AERONET, are you are taking the coarse fraction of the simulated dust fields (i.e. r > 1microm)?

*Dust concentrations from all size bins are taken to calculate the dust AOD as described by Equation 4.*

Page 7 Line 16. You could add the work of Rodríguez, S., Querol, X., Alastuey, A., Kallos, G., and Kakaliagou, O.: Saharan dust contributions to PM10 and TSP levels in Southern and Eastern Spain, Atmos. Environ., 35, 29 2433–2447, 2001.
*Many thanks for the suggestion. The paper is added to the list of references.*

Page 8 Line 2. You could add the work of Basart, S., Pérez, C., Cuevas, E., Baldasano, J. M., and Gobbi, G. P.: Aerosol characterization in Northern Africa, Northeastern Atlantic, Mediterranean Basin and Middle East from direct-sun AERONET observations, Atmos. Chem. Phys., 9, 8265-8282, doi:10.5194/acp-9-8265-2009, 2009.
*Many thanks for the suggestion. We have added the publication to the list of references.*

Page 8 Line 10. For the Eastern Mediterranean region, you should consider the Middle East desert dust sources particularly in spring.
*The focus of the present study is on North African dust transport toward the western Mediterranean basin. Discussing the contribution of dust originating from the Middle East is an interesting topic, no doubt, but beyond the scope of the manuscript. The model setup used here accounts for dust originating from North African dust sources only.*

Page 8 Line 16. Could you quantify "significantly stronger"?
*Calculating the correlation between 10m wind speed and wind speed at 850hPa and 700hPa respectively, the fraction of grid cells which represent a correlation coefficient of 0.4 or larger shows a large difference:*
*fraction correl (10m wind, 850hPa wind) > 0.4: 68.47%*
*fraction correl (10m wind, 700hPa wind) > 0.4: 2.71 %*
*The considered domain is chosen as limited by 20-40N, 20W-30E. This information has been added to the manuscript.*

Page 9 Line 13. You should indicated that they are the dust sources predicted by the model.
*This is clarified now.*

Section 5.1. In addition to AERONET comparison, you should include the spatial comparison of your model results using other aerosol observational datasets as MODIS, SEVIRI/AERUS and EARLINET.
*Many thanks for your suggestion. We have added a comparison with MODIS AOD products.*

Figure 1. Include the locations of Sirtra and Biscaia Gulf.
*Bay of Biscay and Gulf of Sidra are now highlighted in Figure 1.*

---

## Author Comment (AC3) · 17 Oct 2016

**Point-by-point reply to the editor's comments**

*The authors would like to thank the Editor for handling the manuscript, the time he spend on the manuscript, and for providing helpful and constructive comments and suggestions. We have considered carefully all comments made; please find our detailed reply (italic) below.*

**Editor's Comments**

I find that the present focus on summer 2013 is a strong limitation of the manuscript. One possible interesting and somewhat expected outcome of such a study would be a discussion about the 2013 campaign period representativeness based on the analysis of a larger data set including several summers. I would therefore recommend that this point is considered during the likely revision of the paper.

*This comments is in line with the comment made be reviewer #1. We clearly agree and the revised version of the manuscript includes an analysis of the period 1979-2015 using ERA-Interim reanalysis fields. For further details see our reply to the comments of reviewer #1, please.*

---

## Editor Decision (ED1)

**Comments for technical corrections of revised ms. acp-2016-497 entitled "North African dust transport toward the western Mediterranean basin: Atmospheric controls on dust source activation and transport pathways during June-July 2013", by K. Schepanski, M. Mallet, B. Heinold and M. Ulrich.**

*By François Dulac, October 30, 2016*

Thank you for the revision of your manuscript and additional material that really adds significant value to the paper. I am pleased to accept you manuscript for publication pending some technical corrections listed below:

-Please complete the abstract with information about the June-July 1979-2015 EOF that is presently not mentioned.

-Page 1, line 12: "predominant" or "predominance of the".

-P. 3, l. 13: "models" (plural).

-P. 3, l. 14: Menut et al., 2016 seems not appropriate here since you are talking about radiative effect of aerosols and that the cited paper only deals with aerosol optical properties. Appropriate reference from these authors concerning aerosol radiative effects is rather Mailler at al. (2016):
Mailler, S., L. Menut, A. G. di Sarra, S. Becagli, T. Di Iorio, B. Bessagnet, R. Briant, P. Formenti, J.-F. Doussin, J. L. Gómez-Amo, M. Mallet, G. Rea, G. Siour, D. M. Sferlazzo, R. Traversi, R. Udisti, and S. Turquety (2016) On the radiative impact of aerosols on photolysis rates: comparison of simulations and observations in the Lampedusa island during the ChArMEx/ADRIMED campaign, Atmos. Chem. Phys., 16, 1219-1244, doi:10.5194/acp-16-1219-2016, 2016.
Since your reply to comments state that you are now referring to Menut et al. (2016) in the introduction, please cite it differently.

-P. 5, l. 15: at end of sentence, please specify "for dust mobilization and transport".

-P. 7, l. 9. please add here reference to Otto et al. (2009) and the relevant statement in your reply to comment of reviewer #1 (p. 4 of your reply).

-P. 12, l.5: referring here to your statement "not shown", I believe you should rather complete fig. 4 with comparable results from the 1979-2015 period.

-P. 24: check doi presently missing in Ritter et al. (1992) and Rodriguez et al. (2001).

-Fig. 3: it should be made clearer that MODIS AOD plots are for total aerosol but MERRA and COSMO-MUSCAT only for dust AOD (check also the text p. 13, line 5 on).

-Fig. 6 : it is missing here that you probably refer to factor 1 of the EOF.

-Fig 11: please homogenize ticks on both axes (e.g., use a label every 10° and a minor tick every 2° in abscissa).

-Figures : please try to enlarge the character size of plot legends (title, axes, and/or inside box), which are hardly visible to me when printed, especially in Figs. 3, 4, 7-11; in addition, rather than duplicating small colour scales in a number of figures (7, 8, 9, 11), having a single 1 enlarged over the two columns would be more readable.